# On the Importance of Embedding Norms in Self-Supervised Learning

Andrew Draganov [1][2]   Sharvaree Vadgama [3]   Sebastian Damrich [4]   Jan Niklas Böhm [4]   Lucas Maes [5]
Dmitry Kobak[*][4]   Erik Bekkers[*][3]

## Abstract

Self-supervised learning (SSL) allows training data representations without a supervised signal and has become an important paradigm in machine learning. Most SSL methods employ the cosine similarity between embedding vectors and hence effectively embed data on a hypersphere. While this seemingly implies that embedding norms cannot play any role in SSL, a few recent works have suggested that embedding norms have properties related to network convergence and confidence. In this paper, we resolve this apparent contradiction and systematically establish the embedding norm's role in SSL training. Using theoretical analysis, simulations, and experiments, we show that embedding norms (i) govern SSL convergence rates and (ii) encode network confidence, with smaller norms corresponding to unexpected samples. Additionally, we show that manipulating embedding norms can have large effects on convergence speed. Our findings demonstrate that SSL embedding norms are integral to understanding and optimizing network behavior.

## 1. Introduction

Self-supervised learning (SSL) has emerged as a powerful tool for learning representations from unlabeled data. This is because, when trained on large unlabeled datasets with contrastive objectives, SSL models often achieve performance levels comparable to those of supervised methods and can even induce emergent properties (Chen et al., 2020a; Caron et al., 2021). SSL has also substantially advanced multi-modal learning, particularly in vision-language mod-

els (Girdhar et al., 2023; Bommasani et al., 2021). Indeed, prominent methods such as CLIP (Radford et al., 2021), ALIGN (Jia et al., 2021) and Florence (Yuan et al., 2021) all rely on standard SSL objectives discussed in this paper.

These algorithms work by representing similar inputs near each other and dissimilar inputs far apart in an embedding space normalized to a hypersphere. This is done by optimizing objective functions based on the cosine similarity between embeddings: popular SSL frameworks like SimCLR (Chen et al., 2020a) and MoCo (He et al., 2020) optimize the InfoNCE objective, while SimSiam (Chen & He, 2021) and BYOL (Grill et al., 2020) optimize the cosine similarity directly. Thus, theoretical studies of SSL representations only consider the distribution of points on this latent hypersphere (Wang & Isola, 2020; Zimmermann et al., 2021).

At the same time, there has been some empirical evidence that the embedding norms prior to normalization contain meaningful information. For example, Kirchhof et al. (2022) noted that embedding norms are related to network confidence. No theoretical explanation of this phenomenon was given. Separately, Zhang et al. (2020) and Wang et al. (2017) showed that training based on the cosine similarity loss affects the embedding norms and that the embedding norms affect the gradient magnitudes. However, the implications of this for SSL methods have not been fully explored. This implies a clear gap in the literature: several works suggest interactions between SSL models and the embedding norms but the extent of this relationship is unknown. Our work provides the first thorough analysis showing how the embedding norms interact with SSL training dynamics.

**First,** we prove theoretical bounds showing that the embedding norms impose a quadratic slowdown on SSL convergence and verify that this occurs in simulation and in practice. At the same time, we show that optimizing the cosine similarity grows these embedding norms across SSL methods. This leads to a catch-22: small embedding norms are required to train SSL models but these norms grow during training. Put simply, *effectively training SSL models requires managing the embedding norms*. We offer and evaluate several mechanisms for doing this.

**Second,** we argue that, because the embeddings grow with each gradient update, their norms naturally correspond to

[*]Equal contribution  [1]Department of Computer Science, Aarhus University, Denmark  [2]IAS-8, Jülich Forschungszentrum, Germany  [3]AMLab, University of Amsterdam, The Netherlands  [4]Hertie Institute for AI in Brain Health, University of Tübingen, Germany  [5]Mila, Quebec AI Institute, Canada. Correspondence to: Andrew Draganov <draganovandrew@gmail.com>.

*Proceedings of the 42nd International Conference on Machine Learning*, Vancouver, Canada. PMLR 267, 2025. Copyright 2025 by the author(s).

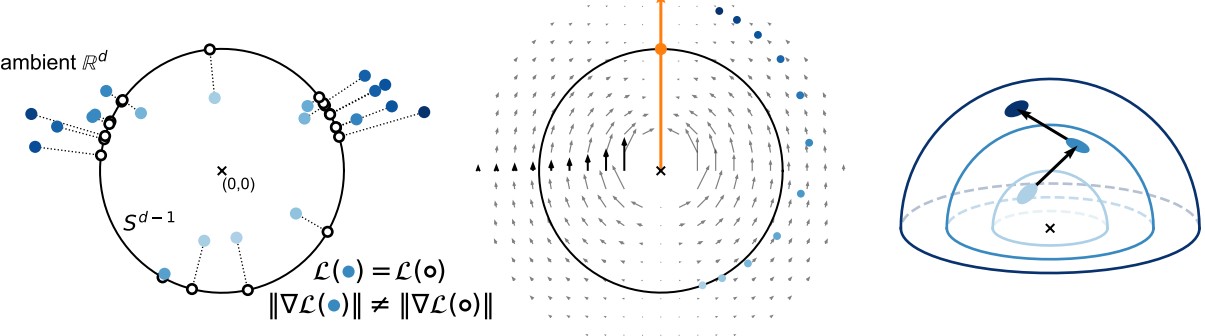

Figure 1. *Left*: Network outputs (blue) lie in ambient $\mathbb{R}^d$, but only their projections (white) onto the hypersphere $S^{d-1}$ enter the loss. However, the embedding norms (shades of blue) influence the norms of the SSL loss gradient. *Middle*: Gradient field of the cosine similarity with respect to the orange direction. Highlighted gradients (black) illustrate the inverse relationship between the gradient and embedding norm (Proposition 3.1). The blue points trace an embedding's trajectory using gradient descent. *Right*: Because gradient updates are orthogonal to an embedding point, they must increase the point's norm (Corollary 3.3).

the frequency of observed latent features. Since models are more certain in frequently observed data (Zhong et al., 2021), we contend that *SSL embedding norms encode a model's confidence*. We show that it consistently holds across SSL methods.

Moreover, these two phenomena interact during SSL training. For instance, since norms both encode uncertainty and scale gradients, SSL models implicitly learn unevenly over the samples — an effect we show must be mitigated when training on imbalanced datasets. Throughout our paper, we draw attention to similar interactions between embedding norms and SSL dynamics by raising several open questions as directions for future work.

This paper supersedes our earlier workshop paper (Draganov et al., 2024). Our code is available at https://github.com/Andrew-Draganov/SSLEmbeddingNorms.

## 2. Preliminaries and Related Work

Although this paper concerns general SSL techniques, we will use the familiar language of image representation learning. The typical SSL pipeline consists of obtaining two augmented variants $x_i$ and $x_j$ from an input image $x$ and ensuring that the corresponding embeddings $z_i$ and $z_j$ have high cosine similarity. We refer to $(x_i, x_j)$ (resp. $(z_i, z_j)$) as 'positive' pairs of points (resp. embeddings). Since attracting all the points together will collapse the representation, each SSL method has an additional mechanism to prevent this. The most common approach enforces low similarity between embeddings of dissimilar inputs: for unrelated inputs $x_i$ and $x_k$, the embeddings $z_i$ and $z_k$ should be far apart.

### 2.1. Preliminaries & SSL methods

Perhaps the prototypical method in this family is Sim-CLR (Chen et al., 2020a), which appears among a broader line of *contrastive* methods for self-supervised learning on images. These all utilize repulsions from dissimilar (negative) samples to prevent representation collapse (Oord et al., 2018; Henaff, 2020; He et al., 2020; Chen et al., 2020b; Misra & Maaten, 2020). In practice, this is performed by optimizing the InfoNCE loss, which we write with respect to embedding $z_i$ as

$$\mathcal{L}_{ij}(\mathbf{Z}) = -\log \frac{\text{ExpSim}(z_i, z_j)}{\sum_{k \neq i} \text{ExpSim}(z_i, z_k)} \quad (1)$$
$$= -\underbrace{\hat{z}_i^\top \hat{z}_j / \tau}_{\mathcal{L}_{ij}^A(\mathbf{Z})} + \underbrace{\log(S_i)}_{\mathcal{L}_{ij}^R(\mathbf{Z})},$$

where $\hat{z} = z/\|z\|$ is the unit vector of $z$, $\text{ExpSim}(a, b) = \exp\left(\hat{a}^\top \hat{b} / \tau\right)$ is the exponent of the temperature-scaled cosine similarity, and $S_i = \sum_{k \neq i} \text{ExpSim}(z_i, z_k)$ represents the sum over all $k \neq i$ in the batch. For ease of interpretation, we split the loss term into attractive and repulsive components, resp. $\mathcal{L}_{ij}^A(\mathbf{Z})$ and $\mathcal{L}_{ij}^R(\mathbf{Z})$. Note that $\mathcal{L}_{ij}^A(\mathbf{Z})$ is simply the negative cosine similarity between positive pair $z_i$ and $z_j$ scaled by $\tau$. We assume $\tau = 1$ for simplicity; in SimCLR, $\tau = 0.5$.

Another common contrastive objective function is the triplet loss, wherein one normalizes the embeddings to the hypersphere and minimizes the mean squared error between positive samples while maximizing the mean squared error between negative ones (Schroff et al., 2015). Due to the normalization, this implicitly optimizes the cosine similarity

between embeddings (Grill et al., 2020).

Curiously, methods like *BYOL* (Grill et al., 2020) and *Sim-Siam* (Chen & He, 2021) showed that one can simply optimize $\mathcal{L}_{ij}^{A}$, and avoid collapse by applying only the gradients to embedding $z_i$ (rather than to both $z_i$ and $z_j$). We refer to these as *non-contrastive* methods. Throughout this paper, we will use SimCLR with the InfoNCE loss to represent a prototypical contrastive SSL model and SimSiam with the negative cosine similarity to represent a prototypical non-contrastive SSL model. As is standard (Oquab et al., 2023), we use the $k$NN classifier accuracy on the embedding space to evaluate the quality of a learned representation. We provide linear probe results in Table S3 in the Appendix to show the generality of our results.

### 2.2. Related Work

**Analysis of SSL Embeddings**   Due to its reliance on the cosine similarity, the seminal work of Wang & Isola (2020) showed that contrastive learning must satisfy two requirements: all positive pairs must be near one another (alignment) and all negative samples must spread evenly over the hypersphere (uniformity). Expanding on this blueprint, subsequent works have sought to formalize the learning capacity of contrastive methods (Saunshi et al., 2019; Zimmermann et al., 2021; HaoChen et al., 2021; Saunshi et al., 2022) while much of the research into non-contrastive methods has focused on how their architectures help to prevent collapse (Jing et al., 2021; Tian et al., 2021a; Zhang et al., 2022; Richemond et al., 2023).

**SSL on Imbalanced Data**   It is an active area of research to understand how contrastive learning performs over various data distributions (Zimmermann et al., 2021). Although Liu et al. (2022) showed that SSL training is relatively robust to imbalanced classes, there are mechanisms which can improve its performance (Tian et al., 2021b; Van Assel & Balestriero, 2024). Nonetheless, foundation models which use contrastive learning require balancing the data distribution before training (Xu et al., 2023).

**Relationship of Embedding Norms and SSL**   While SSL representation learning has largely focused on hyperspherical embedding distributions, some work has examined the embedding norm's role. Wang et al. (2017); Zhang et al. (2018; 2020) all noted that embedding norms inversely scale gradients under the cosine similarity loss and suggested that the embeddings must grow. However, each paper largely brushed this interaction away: Wang et al. (2017) suggested that $\ell_2$-normalization suffices to handle the embedding norms, Zhang et al. (2018) states that attempts to resolve this interaction were unsuccessful and Zhang et al. (2020) provides a regularization term which shrinks the *variance* of the embedding norms but leaves

their average magnitude unmanaged. Our paper therefore differs from the prior literature by (i) extending these results to the InfoNCE loss, (ii) evaluating the full extent to which large embedding norms affect real-world training, and (iii) providing principled mechanisms for addressing this effect.

Similarly, it has been suggested (Scott et al., 2021; Kirchhof et al., 2022; 2023b) that an SSL embedding's magnitude could serve as a measure for the model's certainty. This was evidenced in two ways: first, by qualitatively showing that samples with high-embedding norm are good representatives of the classes (see Figure S6) and, second, by finding that the embedding norm is smaller for out-of-distribution (OOD) samples. Importantly, these references only *show* the existence of this relationship but, to our knowledge, there has not been an explanation provided for why this occurs nor a systematic evaluation of the settings in which norms encode model certainty. Thus, our work expands the literature by establishing how and why embedding norms encode SSL model confidence and providing a thorough empirical analysis for the phenomenon.

## 3. The Properties of SSL Gradients

We begin by studying the gradients of the cosine similarity with respect to an arbitrary point $z_i$. Throughout this section, we assume $\tau = 1$ and refer to $\mathbf{Z}$ as any set of points in $\mathbb{R}^d$, with no other assumptions over the distribution. The gradient acting on one of these points has the following structure:

**Proposition 3.1.** *[Prop. 3 in Zhang et al. (2020);[1] proof in A.1] Let $\mathbf{Z}$ be a set of points in $\mathbb{R}^d$ and let $z_i$ and $z_j$ be a positive pair in $\mathbf{Z}$. Let $\phi_{ij}$ be the angle between $z_i$ and $z_j$. Then the gradient of $\mathcal{L}_{ij}^{A}(\mathbf{Z})$ with respect to $z_i$ is*

$$\nabla_i^{A} = -\frac{1}{\|z_i\|} \left( \mathbf{I}_d - \frac{z_i z_i^{\top}}{\|z_i\|^2} \right) \frac{z_j}{\|z_j\|} = -\left( \frac{\hat{z}_j}{\|z_i\|} \right)_{\perp z_i}$$

*where $a_{\perp b}$ is the component of $a$ orthogonal to $b$. This has magnitude $\|\nabla_i^{A}\| = \frac{\sin(\phi_{ij})}{\|z_i\|}$.*

This has an easy interpretation: $\mathbf{I}_d - \frac{z_i z_i^{\top}}{\|z_i\|^2}$ projects the unit vector $\hat{z}_j$ onto the subspace orthogonal to $z_i$. This projected vector is then inversely scaled by $\|z_i\|$. We visualize this in Figure 1. A similar result holds for the InfoNCE loss:

**Proposition 3.2.** *[Proof in A.2] Let $\mathbf{Z}$ be a set of points in $\mathbb{R}^d$, $z_i$ and $z_j$ be a positive pair in $\mathbf{Z}$, and $\nabla_i^{A}$ be as in Prop. 3.1. Then the gradient of $\mathcal{L}_{ij}(\mathbf{Z})$ with respect to $z_i$ is*

$$\nabla_i = \nabla_i^{A} + \frac{1}{\|z_i\|} \cdot \sum_{k \neq i} \left( \hat{z}_k \cdot \frac{\text{ExpSim}(z_i, z_k)}{S_i} \right)_{\perp z_i} . \quad (2)$$

---

[1] Zhang et al. (2020) also showed corresponding results under SGD with momentum and Adam optimization (Kingma, 2014).

Let $z_l$, $l \neq i$, be a sample in the denominator of $\mathcal{L}_{ij}^{\mathcal{R}}(\mathbf{Z})$, then the gradient of $\mathcal{L}_{ij}(\mathbf{Z})$ with respect to $z_l$ is

$$\nabla_l = -\frac{1}{\|z_l\|} \cdot \frac{\text{ExpSim}(z_i, z_l)}{S_i}(\hat{z}_i)_{\perp z_l}. \qquad (3)$$

In essence, because the InfoNCE loss is a function of the cosine similarity, the chain rule implies that its gradients behave similarly to the cosine similarity's. Specifically, just like those of $\mathcal{L}_{ij}^{\mathcal{A}}$, the gradients of $\mathcal{L}_{ij}^{\mathcal{R}}$ have the properties that (1) they are inversely scaled by $\|z_i\|$ and (2) they exist in $z_i$'s tangent space. Since the InfoNCE loss is the sum of $\mathcal{L}_{ij}^{\mathcal{A}}$ and $\mathcal{L}_{ij}^{\mathcal{R}}$, these properties all extend to the InfoNCE loss as well. Going forward, we refer to any loss function or SSL model as *cosine-similarity-based* (cos.sim.-based) if it exhibits these two properties. This orthogonality has also been noted in Wang et al. (2017) and Zhang et al. (2018).

As a direct consequence of this projection onto the tangent plane, applying the cosine similarity or InfoNCE gradients to a point *must grow its magnitude* (visualized in Figure 1, middle and right):

**Corollary 3.3** (First identified in Wang et al. (2017); Proof in A.3). *Let $z \in \mathbb{R}^d$ and let $z'$ be the result of applying a step of gradient descent with respect to the cosine similarity (Prop. 3.1) or InfoNCE (Prop. 3.2) to $z$. Then $\|z'\| \geq \|z\|$.*

The results in Propositions 3.1, 3.2 and Corollary 3.3 reveal an inevitable catch-22 for self-supervised learning: we require small embeddings to avoid vanishing gradients but optimizing SSL loss functions grows the embeddings. We refer to this as the *embedding-norm effect*. This effect also holds for the mean squared error between normalized embeddings and, by extension, the triplet loss.

Furthermore, Proposition 3.1 has direct implications for convergence rates under the cosine similarity. The gradient's magnitude directly scales the learning rate, since $z_i' = z_i + \gamma \nabla_i = z_i + (\gamma \cdot \|\nabla_i\|)\hat{\nabla}_i$. Thus, Proposition 3.1 can be interpreted as saying that the embedding norm and the sine of the angle parameterize the model's learning rate. Indeed, both quadratically slow down convergence:

**Theorem 3.4** (Proof in A.4). *Let $z_i$ and $z_j$ be embeddings with equal norm, i.e. $\|z_i\| = \|z_j\| = \rho$. Let $z_i' = z_i + \frac{\gamma}{\rho}(z_j)_{\perp z_i}$ and $z_j' = z_j + \frac{\gamma}{\rho}(z_i)_{\perp z_j}$ be the embeddings after maximizing the cosine similarity via a step of gradient descent with learning rate $\gamma$. Then the change in cosine similarity is bounded from above by:*

$$\hat{z}_i'^\top \hat{z}_j' - \hat{z}_i^\top \hat{z}_j < \frac{2\gamma \sin^2 \phi_{ij}}{\rho^2}. \qquad (4)$$

Put simply, the change in the cosine similarity via a step of gradient descent scales quadratically with the embedding's norm and the sine of the angle to its positive counterpart.

## 4. Simulations

We now present a suite of simulations which allow us to characterize how the parameters in Section 3 influence SSL training under idealized conditions. Full implementation and experiment details can be found in Appendix B.

### 4.1. Effect of the Embedding Norms on SSL Training

We start by evaluating to what extent the embedding norms and angles between positive samples slow down convergence. Specifically, we sampled 500 pairs of points directly on $\mathbb{S}^{20}$. We produce many such sets of samples while varying their mean embedding norms and $\phi_{ij}$ values. We then evaluate Theorem 3.4 by applying the cosine similarity gradients to all positive pairs of embeddings until convergence.

Figure 2a plots the number of steps until convergence and shows that, although the convergence rate depends on both parameters, having large embedding norms is *significantly* worse for optimization than having large angles between positive pairs. In essence, the embedding norm's unbounded nature allows it to induce arbitrarily large slowdowns. Meanwhile, the angle between positive samples only has a non-negligible impact as the angle approaches its upper limit $\pi$. Because it is exponentially unlikely for the angle of *every* positive pair to be close to $\pi$, we ignore the angular component of Theorem 3.4 for the remainder of this paper and relegate its further discussion to Appendix C.2.

To further investigate the interaction between weight decay and the embedding-norm effect described in Theorem 3.4, we extend this simulation by augmenting the objective with a regularization term on the embedding norm. That is, we now optimize $\mathcal{L}_{ij} = -(z^i)^\top z^j + \gamma \cdot \|z_i\|^2$, where $\gamma$ represents the weight decay strength. We initialize embedding pairs with different initial norms (1, 4, and 7) to systematically test the quadratic dependence predicted by our theory. The results, shown in Table 1, demonstrate that while weight decay can indeed accelerate convergence by controlling embedding growth, the fundamental quadratic dependence on initial embedding norm persists across all tested regularization strengths. For moderate weight decay values ($0.5 \leq \gamma \leq 1$), we observe faster convergence compared to the unregularized case, with higher regularization providing greater speedup. However, when weight decay becomes too aggressive ($\gamma = 10$), the representation collapses entirely, presumably because the embedding is more incentivized to shrink to the origin than to maximize the cosine similarity. These findings align with our practical experiments in Section 6, where excessive weight decay led to representation collapse.

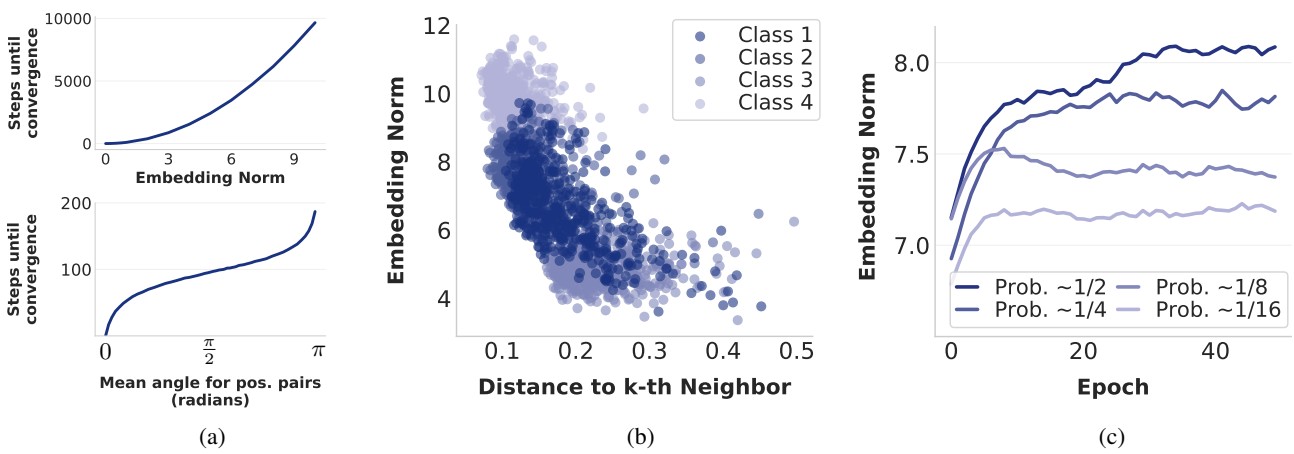

*Figure 2.* Simulations studying the relationship between SSL training and embedding norms. *Left*: Applying cosine similarity gradients to pairs of points converges slower as a function of the points' norm and the sin of their angle. *Middle*: Training via InfoNCE to reconstruct latent classes induces higher norms in dense output regions. *Right*: Training via InfoNCE leads to larger norm for high-frequency classes.

*Table 1.* Steps to convergence for different initial embedding norms under varying weight decay strengths. The objective is modified to $\mathcal{L}_{ij} = -(z^i)^\top z^j + \gamma \cdot \|z_i\|^2$ to simulate weight decay effects.

| Initial Embedding Norm | $\gamma = 0.5$ (Steps) | $\gamma = 1.0$ (Steps) | $\gamma = 10.0$ (Steps) |
|---|---|---|---|
| 1 | 64 | 49 | divergence |
| 4 | 526 | 318 | divergence |
| 7 | 1080 | 610 | divergence |

### 4.2. Effect of SSL Training on the Embedding Norms

We now consider how SSL training affects the embedding norms via a simplified training setting where the data is generated from latent classes. Inspired by Zimmermann et al. (2021) and Kirchhof et al. (2023a), consider an SSL dataset as a set of latent class distributions $\{\tilde{\mathcal{Z}}_1, ..., \tilde{\mathcal{Z}}_k\}$, where each $\tilde{\mathcal{Z}}_i$ is a probability distribution on the $d$-dimensional hypersphere $\mathbb{S}^d$. Let the observations $x \in \mathcal{X} \subset \mathbb{R}^D$ be obtained via a generating process $g : \mathbb{S}^d \to \mathbb{R}^D$. That is, our dataset is obtained by randomly choosing a distribution $\tilde{\mathcal{Z}}_i$, drawing a sample $\tilde{z}$ from it, and applying $g$ to $\tilde{z}$. We are training a neural network $f : \mathcal{X} \to \mathbb{S}^d$ via contrastive learning to produce a learned latent embedding $f(\mathcal{X})$.

We analyze the relationship of parameterized SSL training to the embedding norms by simulating the above scenario. Specifically, we choose centers for 4 latent classes uniformly at random from $\mathbb{S}^{10}$. We then sample 4K points around these centers and normalize them to the hypersphere. From this, we produce the dataset via generating process $g : \mathbb{S}^{10} \subset \mathbb{R}^{11} \to \mathbb{R}^{64}$, where $g$ is given by multiplication by a random matrix. We finally train a 2-layer feedforward

network with the supervised InfoNCE loss function (which explicitly samples positive pairs belonging to the same class) on this dataset.

Figure 2b plots each embedding's magnitude in the learned space as a function of (inverse) density in embedding space. We use the distance to an embedding's $10^{\text{th}}$ nearest neighbor under the cosine similarity metric as a proxy for inverse density. We see that embeddings in dense regions of the embedding space tend to have higher norm. This follows from Corollary 3.3: dense regions of the embedding space receive the most gradient updates and, consequently, those embeddings will grow the most. We also modify the simulation for Figure 2c by providing a class imbalance parameter to the class distribution. Namely, class 1 now has sample probability ~1/2, class 2 has sample probability ~1/4, and so on. We then see that over the course of training, the mean embedding norms for frequent classes grow to higher values than those for sparse classes.

**Takeaway** Across these experiments, samples which are seen more often have higher embedding norm under the InfoNCE loss. This can occur either due to the network considering these samples to be prototypical (and therefore embedding them in dense regions of the learned space) or being otherwise over-represented in the dataset. We point out that these are precisely the settings in which we expect a network to be confident in the embedding. We leave a formal quantitative analysis as an open question:

**Open Question 4.1.** *What theoretical bounds can be made regarding a sample's embedding norm and (a) the accuracy with which it is classified or (b) the corresponding input's dissimilarity from the training data?*

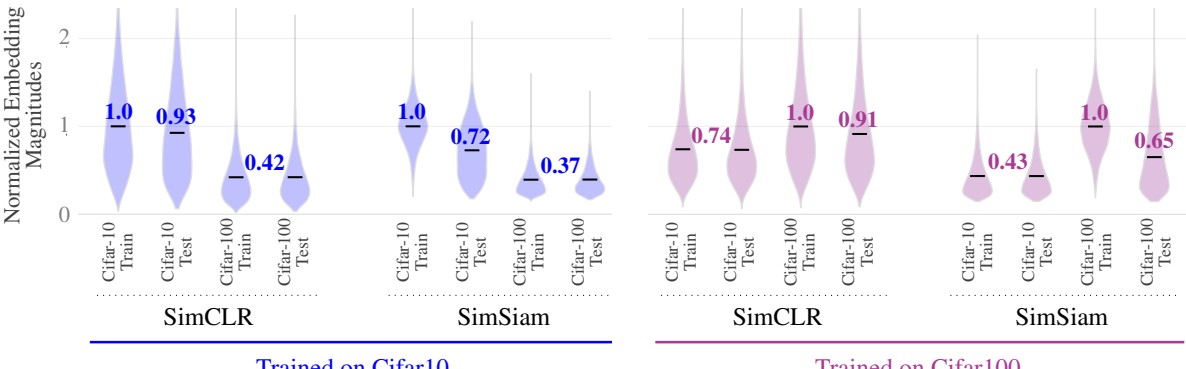

*Figure 3.* Violin plot showing the distribution of embedding norms on each dataset split as a function of which dataset the model was trained on. All values are normalized by the training set's mean embedding magnitude. Black bars represent and are labeled by the mean value of each violin. We use the same augmentations for the train and test sets for consistency.

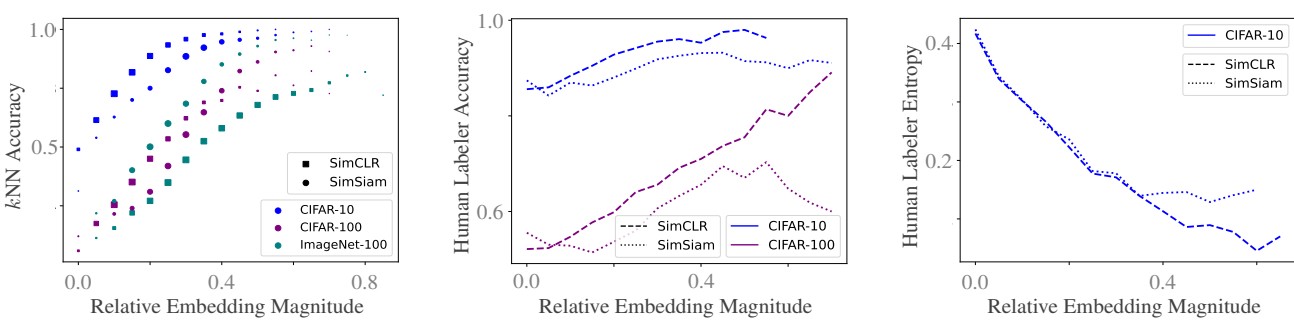

*Figure 4.* Evaluation of how embedding norms correspond to measures of network confidence. In all cases, we normalize the embedding magnitudes by the in-dataset maximum and bin them into twenty buckets. For every bucket with more than $50$ embeddings, we evaluate the corresponding metric. We represent the number of embeddings per bucket using the marker size in the left-hand plot.

## 5. Embedding Norm as Network Confidence

Given the simulations in Section 4.2, we expect that training cos.sim.-based SSL models should result in common input samples receiving higher norm. We therefore use this section to show the various ways in which embedding norms encode a network's confidence in practice. All model implementation details can be found in Appendix C.1.

**Embedding Norms Encode Novelty** Figure 3 demonstrates how embedding norms characterize a sample's "out-of-distributionness". On the left side of the figure, we trained SimCLR and SimSiam models on the Cifar-10 train set for 512 epochs and then compared the embedding norms across different data splits, normalizing all embedding norms by the Cifar-10 train set mean. The results reveal a clear pattern: embedding norms decrease progressively with increasing distributional distance from the training data. For example, the Cifar-10 test set contains novel but distributionally similar samples and therefore results in only slightly reduced norms. In contrast, the Cifar-100 data splits

exhibit substantially smaller norms due to their greater distributional shift. This relationship holds symmetrically when training on Cifar-100 and evaluating on Cifar-10, as seen on the right side of Figure 3.

**Embedding Norms Encode Classification Accuracy** Another measure of a network's confidence in an embedding is the accuracy with which that sample is classified. To this end, we use the same experimental setup as above and train SimCLR and SimSiam on the Cifar-10, Cifar-100, and ImageNet100 datasets.[2] We then normalize the embedding magnitudes by the maximum across the dataset and bucket the embeddings into ranges of $0.05$, giving us 20 embedding buckets over the dataset. Figure 4 (left) then shows the per-bucket accuracy of a $k$NN classifier which was fit on all the embeddings with respect to the cosine similarity metric. Indeed, we see that the $k$NN classifier's accuracy shows a clear monotonic trend with the embedding norms

---

[2]We default to the ImageNet-100 split (Wang & Isola, 2020) at `huggingface.co/datasets/clane9/imagenet-100`.

across datasets and SSL models.

**Embedding Norms Encode Human Confidence** Interestingly, not only does the embedding norm provide a measure for the sample's novelty and its classification accuracy, but it also provides a signal for human labelers' confidence and their agreement among one another. Using the Cifar-10-N and Cifar-100-N labels from Wei et al. (2021), where each training sample is labeled by the consensus label over multiple human annotators, Figure 4 (middle) shows higher embedding norms correspond to more accurate consensus labels. Similarly, the Cifar-10-H dataset from Peterson et al. (2019) provides ∼50 human predictions for each image from the Cifar-10 test set, allowing us to evaluate the entropy of the label distribution. Figure 4 (right) shows that, as the embedding norms grow, the human labels have less entropy (i.e., they are more likely to agree with one another).

**Takeaways** Under the common assumption that an SSL embedding's direction represents its information, these experiments show that the embedding's norm represents how *confident* the network was in this information. Furthermore, this measure of network confidence is inherent to all cos.sim.-based loss functions and emerges naturally during training. Thus, an SSL latent space looks less like a smooth sphere and more like a spiky ball, with the spikes corresponding to regions of known data samples. This observation has implications for few-shot learning settings, in which one has pre-trained on a large dataset and then wishes to adjust the model to a second, smaller dataset:

**Open Question 5.1.** *By using the embedding norm as a measure for a sample's novelty, can one more precisely guide the training process on unseen inputs?*

## 6. The Embedding-Norm Effect in Practice

To understand how the embedding-norm effect influences cosine-similarity-based SSL training, we investigate three distinct interventions which should mitigate it. These interventions provide controlled settings to analyze the empirical relationship between embedding norms and SSL training. Our experiments are on the Cifar-10, Cifar-100, Imagenet-100 and Tiny-Imagenet (Le & Yang, 2015) datasets.

**Weight Decay** Our first intervention mechanism — weight decay — is already present in essentially all SSL models. The idea here is that adding a penalty on the weights implicitly regularizes the embedding norms (Wang et al., 2017). Figure 5 demonstrates this effect in SimCLR and SimSiam training: without weight decay ($\gamma = 0$), embeddings grow unconstrained, while excessive weight decay ($\gamma = 5 \cdot 10^{-2}$) causes collapse. With appropriate values ($\gamma = 10^{-5}$ for SimCLR, $\gamma = 5 \cdot 10^{-4}$ for SimSiam), norms decrease gradually, leading to improved $k$NN accuracy. Interestingly, the

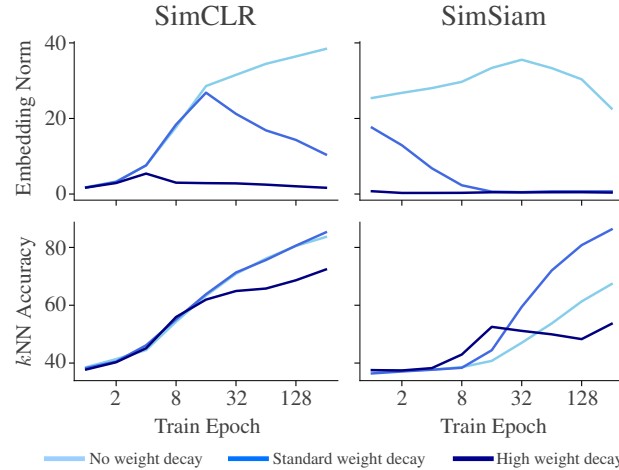

*Figure 5.* Effect of the weight decay on the embedding norms in SimCLR and SimSiam. Epochs are log-scale and go from 1 to 256. Corresponding $k$NN accuracies are plotted in the bottom row.

```python
@torch.no_grad()
def cut_init(model, c):
    for param in model.parameters():
        param.data = param.data / c
```

*Listing 1.* PyTorch code for our cut-initialization layer.

embedding norm's impact on the $k$NN accuracy is much more pronounced in the attraction-only setting. We also see that, even without weight decay, the embeddings can shrink (as occurs for SimSiam). We attribute this to the embeddings being produced by shared weights: although the gradients require all embeddings to grow, a shared weights matrix may not be able to independently update every point's position.

**Cut-Initialization** As shown in Figure 5, weight decay gradually reduces embedding norms over time, but doesn't control them at the start of training. To address this, we propose *cut-initialization* — dividing all network weights by a constant $c$ at initialization. This ensures small embedding norms at initialization which are then kept small via weight decay. We implement this uniformly across all layers for simplicity (Listing 1). Interestingly, a variant of this can be found in HuggingFace's default image transformer code (PyTorch Foundation, 2025) and we know at least one cos.sim.-based SSL model which uses it (Peng et al., 2022).

We study the interplay between cut-initialization and weight decay values on SimCLR and SimSiam in Table 2. Specifically, we report the $k$NN classification accuracy after 100 epochs on the Cifar-100 and ImageNet-100 datasets and see that, in both the contrastive and non-contrastive settings, pairing cut-initialization with weight decay accelerates the training process. As was the case for the weight decay, the

*Table 2.* $k$NN accuracy at epoch 100 for various values of cut-constant $c$ and weight decay $\lambda$, color-coded by value. *Left*: SimCLR on Cifar-100. *Right*: SimSiam on ImageNet-100. Default weight-decay is underlined.

| | | SimCLR | | | | SimSiam | | |
|---|---|---|---|---|---|---|---|---|
| | | Weight Decay $\lambda$ | | | | Weight Decay $\lambda$ | | |
| | | 1e-8 | 1e-6 | 5e-6 | 1e-5 | 5e-5 | 1e-4 | 5e-4 | 1e-3 |
| | $c=1$ | 40.8 | 40.5 | 40.9 | 41.5 | 36.7 | 38.5 | 40.5 | 44.7 |
| Cut | $c=2$ | 42.7 | 42.8 | 42.9 | 42.2 | 41.1 | 44.0 | 48.8 | 42.1 |
| | $c=4$ | 42.3 | 41.4 | 42.0 | 41.1 | 40.2 | 41.2 | 49.8 | 49.1 |
| | $c=8$ | 37.1 | 36.8 | 37.9 | 37.3 | 44.4 | 46.4 | 50.2 | 53.2 |

*Table 3.* Imagenet-100 $k$NN accuracy (epoch 100) for MoCo/Dino.

| | | Top-1 | Top-5 |
|---|---|---|---|
| MoCo | $c=1$ | 38.7 | 69.7 |
| V2 | $c=3$ | 43.8 | 71.8 |
| MoCo | $c=1$ | 54.8 | 82.7 |
| V3 | $c=3$ | 58.8 | 86.0 |
| Dino | $c=1$ | 43.7 | 76.6 |
| | $c=3$ | 29.0 | 55.6 |

difference is stronger in the non-contrastive setting. $c=2$ and $c=4$ performed best for SimCLR, providing an additional 1-2% in accuracy at the default weight decay, while $c=8$ led to about a 10% increase for SimSiam. Extrapolating from the trends in Table 2, we use $c=3$ for SimCLR and $c=9$ for SimSiam going forward. These are futher analyzed in Figure S4, which also includes BYOL experiments. We show the $k$NN classifier accuracies at 500 epochs with and without cut-initialization in Table 4. Here we see that pairing SSL models with cut-initialization often helps the model reach higher final accuracies.

We also evaluate cut-initialization in imbalanced data settings. For Cifar-10, we use the exponential split from Van Assel & Balestriero (2024), where class $i$ has $n_i = 5000 \cdot 1.5^{-i}$ samples. Similarly for Cifar-100, the $i$-th class receives $n_i = 500 \cdot 1.034^{-i}$ samples. This way, all classes are represented and both imbalanced datasets contain roughly 15K samples. We also use Flowers102's naturally long-tailed test set for training (Nilsback & Zisserman, 2008), evaluating on its validation set. Table 5 then shows that, in class-imbalanced settings, pairing SSL training with interventions on the embedding-norm effect can provide double-digit accuracy improvements.

Lastly, Table 3 shows that cut-initialization also provides the expected boost in performance for MoCov2 and MoCov3 (Chen et al., 2020b; Fan et al., 2021), which are cos.sim.-based but use a transformer backbone (Dosovitskiy et al., 2020). We point out that Dino's objective function does not use the cosine similarity (Caron et al., 2021). Consequently, it does not benefit from cut-initialization.

**GradScale Layer** Perhaps the cleanest way to overcome the embedding-norm effect is to simply rescale the gradient directly. We achieve this by introducing a custom PyTorch `autograd.Function` which we refer to as *GradScale* (for a full implementation, see Listing 2 in the Appendix). This layer accepts a *power* parameter $p$ and is simply the identity function in the forward pass. However, the backwards pass multiplies each sample $z_i$'s contribution to the gradient by $\|z_i\|^p$. Thus, choosing power $p=0$ gives the default setting while choosing $p=1$ will cancel the gradi-

*Table 4.* $k$NN accuracies at epoch 1000 for Cifar-10/100 and at epoch 500 for ImageNet/TinyImagenet for default, cut-initialized and GradScale training on standard image datasets.

| | | Cifar 10 | Cifar 100 | Imagenet 100 | Tiny Imagenet |
|---|---|---|---|---|---|
| | Default | 87.7 | 56.6 | 59.4 | 36.0 |
| SimCLR | Cut ($c=3$) | 88.2 | 60.1 | 60.9 | 37.7 |
| | GradScale | 88.2 | 58.2 | 01.0 | 38.1 |
| SimSiam | Default | 88.1 | 61.8 | 62.0 | 41.4 |
| | Cut ($c=9$) | 88.4 | 62.6 | 67.2 | 41.7 |

ents' dependence on the embedding norm. We visualize the resulting gradient fields with powers $p=0$ and $p=1$ for a 2D embedding space in Figure 6. We refer to training with GradScale power $p=1$ simply as GradScale. This can also be interpreted as a per-point adaptive learning rate that is scaled by each point's embedding norm.

Under GradScale, the gradient norms differ from those during default training, necessitating a new learning rate schedule. Traditionally, SSL models are trained with 10 epochs of linear warmup followed by a cosine-annealing schedule (Chen et al., 2020a). However, the schedule has an implicit division by the embedding norms over the course of training. We therefore simulate this effective learning rate for the GradScale setting. Namely, we choose base learning rate $\gamma' = \gamma/6$ with 100 linear warmup epochs followed by cosine annealing, where $\gamma$ is the default learning rate. This was the first relatively stable schedule which we found for the $p=1$ setting and we performed no additional tuning.

Table 4 demonstrates that, when training remains stable, SimCLR's $k$NN accuracy benefits from GradScale's embedding norm cancellation. Consistent with our cut-initialization experiments, this improvement becomes particularly pronounced on the class-imbalanced datasets in Table 5: GradScale provides a roughly 5% accuracy increase across the imbalanced datasets. While these results are promising, we observed that GradScale with $p=1$ failed to converge on Imagenet-100 and for non-contrastive models. This limitation aligns with these models' known sensitivity issues (Zhang et al., 2022; Richemond et al., 2023).

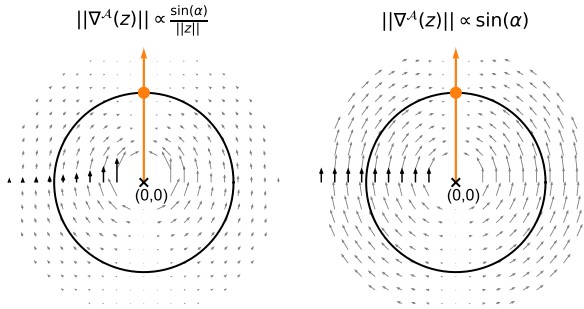

$||\nabla^{\mathcal{A}}(z)|| \propto \frac{\sin(\alpha)}{||z||}$

$||\nabla^{\mathcal{A}}(z)|| \propto \sin(\alpha)$

*Table 6.* $k$NN accuracies at epoch 100 using the Adam optimizer for SimCLR and SimSiam under embedding norm interventions.

|  |  | Cifar-10 | Cifar-100 |
|---|---|---|---|
| SimCLR | Default | 79.6 | 44.3 |
|  | Cut | 79.9 | 45.3 |
|  | GradScale | 80.5 | 46.6 |
| SimSiam | Default | 73.7 | 36.4 |
|  | Cut | 79.8 | 44.9 |

*Figure 6. Left*: the default gradient field of the cosine similarity with respect to the north-pointing line (orange). *Right*: the gradient field using the GradScale layer with $p = 1$.

*Table 5.* $k$NN accuracies at epoch 500 for default, cut-initialized and GradScale training on class-imbalanced image datasets.

|  |  | Cifar-10 Unb. | Cifar-100 Unb. | Flowers Unb. |
|---|---|---|---|---|
| SimCLR | Default | 56.5 | 24.3 | 42.6 |
|  | Cut ($c = 3$) | 61.1 | 26.3 | 61.6 |
|  | GradScale | 61.3 | 29.1 | 47.2 |
| SimSiam | Default | 47.0 | 21.6 | 22.5 |
|  | Cut ($c = 9$) | 61.7 | 31.5 | 39.9 |

**Embedding-norm Effect Under Adaptive Optimizers** A potential concern is that our theoretical analysis, which focuses on standard gradient descent, may not extend to adaptive optimizers. To address this, we evaluate our embedding norm mitigation strategies using the Adam optimizer (Kingma, 2014). As shown in Table 6, the embedding-norm effect persists under Adam optimization: both cut-initialization and GradScale continue to provide consistent improvements across SimCLR and SimSiam on both Cifar-10 and Cifar-100. Table S4 reveals that our mitigation strategies have the expected effects on embedding magnitudes even under adaptive optimization: cut-initialization ensures small norms ($||z|| = 2.1$) while GradScale allows them to grow ($||z|| = 175$) without affecting convergence.

**Takeaways** These results make it clear that the embedding norm effect impacts SSL training — particularly in non-contrastive models — and can be mitigated using our proposed strategies. The effect appears most detrimental in class-imbalanced settings, aligning with our results on SSL confidence: imbalanced data creates variance in embedding norms, destabilizing training. Nonetheless, there remain questions which are beyond the scope of this work:

**Open Question 6.1.** *Why are non-contrastive architectures more sensitive to the embedding-norm effect?*

In addition to seeing that the embedding-norm effect is more pronounced in attraction-only settings, we have found that the embeddings can shrink even in the absence of weight decay. We attribute the latter phenomenon to the network's shared weights: while our theory predicts uniform embedding growth, producing these embeddings via a single set of weights creates competition between different regions of the latent space.

Finally, we have been careful to not describe the embedding-norm effect as a strictly negative phenomenon. Consider the common transfer-learning setting in which prototypical class examples should anchor the learned representation (Pan et al., 2019; Lee et al., 2022). Our findings suggest the embedding-norm effect may naturally support this goal: prototypical examples get large embedding norms and consequently receive smaller gradients.

**Open Question 6.2.** *Are there SSL training schemes in which the embedding-norm effect is beneficial?*

## 7. Discussion

In this work, we investigated a fundamental aspect of cosine similarity-based self-supervised learning: embedding norms serve a dual role, both inversely scaling gradients and encoding model certainty. These characteristics are intrinsic to standard SSL models and our analysis showed how they alter the expected training dynamics in both standard and class-imbalanced settings.

Given that these properties are natively present in widely-used SSL approaches, they suggest several research directions beyond those already stated in the paper. First, embedding norms could serve as simple yet effective reliability metrics during inference. Second, the embedding norms provide a new lens for studying the modality gap in multi-modal representation learning (Liang et al., 2022). Lastly, the interplay between the embedding norm's roles — as both a confidence metric and gradient scalar — suggests training pipelines that explicitly leverage both mechanisms.

## Impact Statement

The goal of our paper is to advance the field of machine learning. We do not see any potential societal consequences of our work that would be worth specifically highlighting.

## Acknowledgements

Andrew Draganov is partially supported by the Independent Research Fund Denmark (DFF) under a Sapere Aude Research Leader grant No 1051-00106B, by a StiboFund IT travel grant for PhD students and by project W2/W3-108 Impuls und Vernetzungsfonds der Helmholtz-Gemeinschaft. Sharvaree Vadgama is supported by the Hybrid Intelligence Center, a 10-year program funded by the Dutch Ministry of Education, Culture and Science through the Netherlands Organisation for Scientific Research (NWO). This work was partially funded by the Gemeinnützige Hertie-Stiftung, the Cyber Valley Research Fund (D.30.28739), and the National Institutes of Health (UM1MH130981). The content is solely the responsibility of the authors and does not necessarily represent the official views of the National Institutes of Health. Dmitry Kobak is a member of the Germany's Excellence cluster 2064 "Machine Learning — New Perspectives for Science" (EXC 390727645). The authors thank the International Max Planck Research School for Intelligent Systems (IMPRS-IS) for supporting Jan Niklas Böhm.

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

# A. Proofs

## A.1. Proof of Proposition 3.1

**Proposition 3.1.** *[Prop. 3 in Zhang et al. (2020);[3] proof in A.1] Let $\mathbf{Z}$ be a set of points in $\mathbb{R}^d$ and let $z_i$ and $z_j$ be a positive pair in $\mathbf{Z}$. Let $\phi_{ij}$ be the angle between $z_i$ and $z_j$. Then the gradient of $\mathcal{L}_{ij}^{\mathcal{A}}(\mathbf{Z})$ with respect to $z_i$ is*

$$\nabla_i^{\mathcal{A}} = -\frac{1}{\|z_i\|}\left(\mathbf{I}_d - \frac{z_i z_i^{\top}}{\|z_i\|^2}\right)\frac{z_j}{\|z_j\|} = -\left(\frac{\hat{z}_j}{\|z_i\|}\right)_{\perp z_i}$$

*where $a_{\perp b}$ is the component of $a$ orthogonal to $b$. This has magnitude $\|\nabla_i^{\mathcal{A}}\| = \frac{\sin(\phi_{ij})}{\|z_i\|}$.*

*Proof.* We are taking the gradient of $\mathcal{L}_i^{\mathcal{A}}$ as a function of $z_i$. The principal idea is that the gradient has a term with direction $\hat{z}_j$ and a term with direction $-\hat{z}_i$. We then disassemble the vector with direction $\hat{z}_j$ into its component parallel to $z_i$ and its component orthogonal to $z_i$. In doing so, we find that the two terms with direction $z_i$ cancel, leaving only the one with direction orthogonal to $z_i$.

Writing it out fully, we have $\mathcal{L}_i^{\mathcal{A}} = -z_i^{\top} z_j/(\|z_i\| \cdot \|z_j\|)$. Taking the gradient amounts to using the quotient rule, with $f = -z_i^{\top} z_j$ and $g = \|z_i\| \cdot \|z_j\| = \sqrt{z_i^{\top} z_i} \cdot \sqrt{z_j^{\top} z_j}$. Taking the derivative of each, we have

$$f' = -\mathbf{z}_j$$
$$g' = \|z_j\|\frac{z_i}{\sqrt{z_i^{\top} z_i}} = \|z_j\|\frac{\mathbf{z}_i}{\|z_i\|}$$
$$\implies \frac{f'g - g'f}{g^2} = \frac{-\left(\mathbf{z}_j \cdot \|z_i\| \cdot \|z_j\|\right) + \left(\|z_j\|\frac{\mathbf{z}_i}{\|z_i\|} \cdot z_i^{\top} z_j\right)}{\|z_i\|^2 \cdot \|z_j\|^2}$$
$$= \frac{-\mathbf{z}_j}{\|z_i\| \cdot \|z_j\|} + \frac{\mathbf{z}_i z_i^{\top} z_j}{\|z_i\|^3 \|z_j\|},$$

where we use boldface $\mathbf{z}$ to emphasize which direction each term acts along. We now substitute $\cos(\phi_{ij}) = z_i^{\top} z_j/(\|z_i\| \cdot \|z_j\|)$ in the second term to get

$$\frac{f'g - g'f}{g^2} = \frac{-\hat{z}_j}{\|z_i\|} + \frac{\mathbf{z}_i \cos(\phi)}{\|z_i\|^2} \tag{5}$$

It remains to separate the first term into its sine and cosine components and perform the resulting cancellations. To do this, we take the projection of $\hat{z}_j = \mathbf{z}_j/\|z_j\|$ onto $\mathbf{z}_i$ and onto the plane orthogonal to $\mathbf{z}_i$. The projection of $\hat{z}_j$ onto $\mathbf{z}_i$ is given by

$$\cos\phi_{ij}\frac{\mathbf{z}_i}{\|z_i\|}$$

while the projection of $\mathbf{z}_j/\|z_j\|$ onto the plane orthogonal to $\mathbf{z}_i$ is

$$\left(\mathbf{I} - \frac{z_i z_i^{\top}}{\|z_i\|^2}\right)\frac{\mathbf{z}_j}{\|z_j\|}.$$

It is easy to assert that these components sum to $\mathbf{z}_j/\|z_j\|$ by replacing the $\cos\phi_{ij}$ by $\frac{z_i^{\top} z_j}{\|z_i\| \cdot \|z_j\|}$.

---

[3]Zhang et al. (2020) also showed corresponding results under SGD with momentum and Adam optimization (Kingma, 2014).

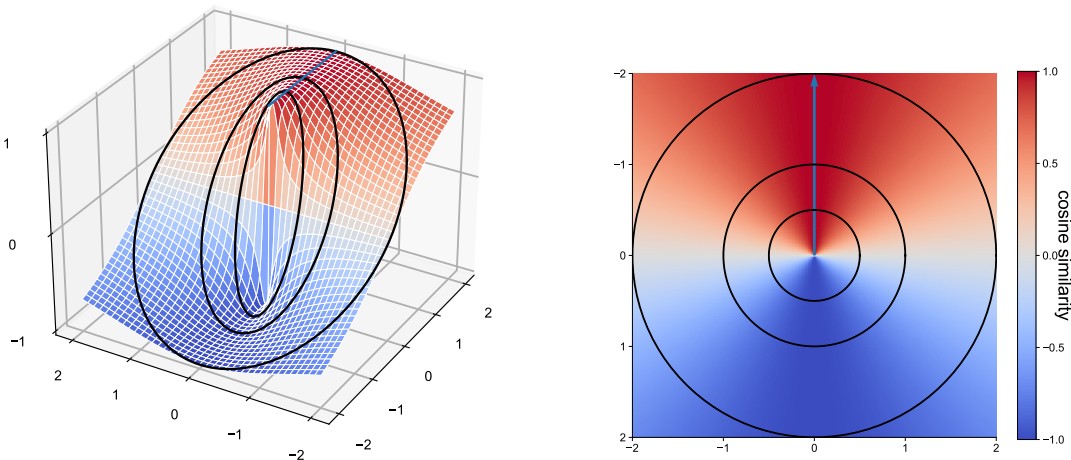

*Figure S1.* Cosine similarity with respect to the direction indicated by the blue line. Three circles of radii 0.5, 1, and 2 are superimposed to show that, for higher norms, the cosine similarity is less steep. Left: 3D Surface plot, right: 2D topview plot.

We plug these into Eq. 5 and cancel the first and third term to arrive at the desired value:

$$
\begin{aligned}
\frac{f'g - g'f}{g^2} =& -\frac{1}{\|z_i\|}\cos\phi\frac{\mathbf{z}_i}{\|z_i\|} \\
& -\frac{1}{\|z_i\|}\cdot\left(\mathbf{I} - \frac{z_i z_i^\top}{\|z_i\|^2}\right)\frac{\mathbf{z}_j}{\|z_j\|} \\
& +\frac{\mathbf{z}_i\cos(\phi)}{\|z_i\|^2} \\
=& \frac{-1}{\|z_i\|}\cdot\left(\mathbf{I} - \frac{z_i z_i^\top}{\|z_i\|^2}\right)\frac{\mathbf{z}_j}{\|z_j\|}.
\end{aligned}
$$

$\square$

We visualize the loss landscape of the cosine similarity function in Figure S1.

### A.2. InfoNCE Gradients

**Proposition 3.2.** *[Proof in A.2] Let $\mathbf{Z}$ be a set of points in $\mathbb{R}^d$, $z_i$ and $z_j$ be a positive pair in $\mathbf{Z}$, and $\nabla_i^{\mathcal{A}}$ be as in Prop. 3.1. Then the gradient of $\mathcal{L}_{ij}(\mathbf{Z})$ with respect to $z_i$ is*

$$
\nabla_i = \nabla_i^{\mathcal{A}} + \frac{1}{\|z_i\|}\cdot\sum_{k\neq i}\left(\hat{z}_k\cdot\frac{\mathrm{ExpSim}(z_i, z_k)}{S_i}\right)_{\perp z_i}. \tag{2}
$$

*Let $z_l$, $l \neq i$, be a sample in the denominator of $\mathcal{L}_{ij}^{\mathcal{R}}(\mathbf{Z})$, then the gradient of $\mathcal{L}_{ij}(\mathbf{Z})$ with respect to $z_l$ is*

$$
\nabla_l = -\frac{1}{\|z_l\|}\cdot\frac{\mathrm{ExpSim}(z_i, z_l)}{S_i}(\hat{z}_i)_{\perp z_l}. \tag{3}
$$

*Proof.* We are interested in the gradient of $\mathcal{L}_i^{\mathcal{R}}$ with respect to $z_i$. By the chain rule, we get

$$\nabla_i^{\mathcal{R}} = -\frac{\sum_{k \neq i} \text{ExpSim}(z_i, z_k) \frac{\partial \frac{z_i^\top z_k}{\|z_i\| \cdot \|z_k\|}}{\partial z_i}}{\sum_{k \neq i} \text{ExpSim}(z_i, z_k)}$$

$$= -\frac{\sum_{k \neq i} \text{ExpSim}(z_i, z_k) \frac{\partial \frac{z_i^\top z_k}{\|z_i\| \cdot \|z_k\|}}{\partial z_i}}{S_i}$$

It remains to substitute the result of Prop. 3.1 for $\partial \frac{z_i^\top z_k}{\|z_i\| \cdot \|z_k\|} / \partial z_i$.

We sum this with the gradients of the attractive term to obtain the full InfoNCE gradient for a positive sample.

Similarly, for $z_l$ we obtain

$$\nabla_l = \nabla_l^{\mathcal{R}} = -\frac{\text{ExpSim}(z_i, z_l)}{\sum_{k \neq i} \text{ExpSim}(z_i, z_k)} \frac{\partial \frac{z_i^\top z_l}{\|z_i\| \cdot \|z_l\|}}{\partial z_l} = -\frac{\text{ExpSim}(z_i, z_l)}{S_i \|z_l\|} (\hat{z}_i)_{\perp z_l}, \tag{6}$$

where the last equality follows again from Prop. 3.1. □

We note that the gradients of the InfoNCE loss on positive and negative samples are always tangent to the respective points.

### A.3. Proof of Corollary 3.3

*Proof.* First, consider that we applied the cosine similarity's gradients from Proposition 3.1. Since $z_i$ and $(z_j)_{\perp z_i}$ are orthogonal, $\|z_i'\|_2^2 = \|z_i\|^2 + \frac{\gamma^2}{\|z_i\|^2} \|(z_j)_{\perp z_i}\|^2$. The second term is positive if $\sin \phi_{ij} > 0$, which is the case save for identical or antipodal points, because then, we have $\phi_{ij} \in (0, \pi)$.

The same exact argument holds for the InfoNCE gradients. The gradient is orthogonal to the embedding, so a step of gradient descent can only increase the embedding's magnitude. □

### A.4. Proof of Theorem 3.4

We first restate the theorem:

Let $z_i$ and $z_j$ be positive embeddings with equal norm, i.e. $\|z_i\| = \|z_j\| = \rho$. Let $z_i'$ and $z_j'$ be the embeddings after 1 step of gradient descent with learning rate $\gamma$. Then the change in cosine similarity is bounded from above by:

$$\hat{z}_i'^\top \hat{z}_j' - \hat{z}_i^\top \hat{z}_j < \frac{\gamma \sin^2 \phi_{ij}}{\rho^2} \left[ 2 - \frac{\gamma \cos \phi}{\rho^2} \right].$$

We now proceed to the proof:

*Proof.* Let $z_i$ and $z_j$ be two embeddings with equal norm[4], i.e. $\|z_i\| = \|z_j\| = \rho$. We then perform a step of gradient descent to maximize $\hat{z}_i^\top \hat{z}_j$. That is, using the gradients in 3.1 and learning rate $\gamma$, we obtain new embeddings $z_i' = z_i + \frac{\gamma}{\|z_i\|} (\hat{z}_j)_{\perp z_i}$ and $z_j' = z_j + \frac{\gamma}{\|z_j\|} (\hat{z}_i)_{\perp z_j}$. Going forward, we write $\delta_{ij} = (\hat{z}_j)_{\perp z_i}$ and $\delta_{ji} = (\hat{z}_i)_{\perp z_j}$, so $z_i' = z_i + \frac{\gamma}{\rho} \delta_{ij}$ and $z_j' = z_j + \frac{\gamma}{\rho} \delta_{ji}$. Notice that since $z_i$ and $\delta_{ij}$ are orthogonal, by the Pythagorean theorem we have $\|z_i'\|^2 = \|z_i\|^2 + \frac{\gamma^2}{\rho^2} \|\delta_{ij}\|^2 \geq \|z_i\|^2$. Lastly, we define $\rho' = \|z_i'\| = \|z_j'\|$.

---

[4]We assume the Euclidean distance for all calculations.

We are interested in analyzing $\hat{z}_i'^\top \hat{z}_j' - \hat{z}_i^\top \hat{z}_j$. To this end, we begin by re-framing $\hat{z}_i'^\top \hat{z}_j'$:

$$
\hat{z}_i'^\top \hat{z}_j' = \left( \frac{z_i + \frac{\gamma}{\rho}\delta_{ij}}{\rho'} \right)^\top \left( \frac{z_j + \frac{\gamma}{\rho}\delta_{ji}}{\rho'} \right)
$$

$$
= \frac{1}{\rho'^2} \left[ z_i^\top z_j + \gamma \frac{z_i^\top \delta_{ji}}{\rho'} + \gamma \frac{z_j^\top \delta_{ij}}{\rho'} + \gamma^2 \frac{\delta_{ij}^\top \delta_{ji}}{\rho'^2} \right].
$$

We now consider that, since $\delta_{ij}$ is the projection of $\hat{z}_j$ onto the subspace orthogonal to $z_i$, we have that the angle between $z_i$ and $\delta_{ji}$ is $\pi/2 - \phi_{ij}$. Plugging this in and simplifying, we obtain

$$
z_i^\top \delta_{ji} = \|z_i\| \cdot \|\delta_{ji}\| \cos(\pi/2 - \phi_{ij})
$$

$$
= \|z_i\| \cdot \|\delta_{ji}\| \sin \phi_{ij}
$$

$$
= \rho \sin^2 \phi_{ij}.
$$

By symmetry, the same must hold for $z_j^\top \delta_{ij}$.

Similarly, we notice that the angle $\psi_{ij}$ between $\delta_{ij}$ and $\delta_{ji}$ is $\psi_{ij} = \pi - \phi_{ij}$. The reason for this is that we must have a quadrilateral whose four internal angles must sum to $2\pi$, i.e. $\psi_{ij} + \phi_{ij} + 2\frac{\pi}{2} = 2\pi$. Thus, we obtain $\delta_{ij}^\top \delta_{ji} = \|\delta_{ij}\| \cdot \|\delta_{ji}\| \cos(\psi) = -\sin^2 \phi_{ij} \cos \phi_{ij}$.

We plug these back into our equation for $\hat{z}_i'^\top \hat{z}_j'$ and simplify:

$$
\hat{z}_i'^\top \hat{z}_j' = \frac{1}{\rho'^2} \left[ z_i^\top z_j + \gamma \frac{z_i^\top \delta_{ji}}{\rho} + \gamma \frac{z_j^\top \delta_{ij}}{\rho} + \gamma^2 \frac{\delta_{ij}^\top \delta_{ji}}{\rho^2} \right]
$$

$$
= \frac{1}{\rho'^2} \left[ z_i^\top z_j + \gamma \frac{\rho \sin^2 \phi_{ij}}{\rho} + \gamma \frac{\rho \sin^2 \phi_{ij}}{\rho} - \gamma^2 \frac{\sin^2 \phi_{ij} \cos \phi_{ij}}{\rho^2} \right]
$$

$$
= \frac{1}{\rho'^2} \left[ z_i^\top z_j + 2\gamma \sin^2 \phi_{ij} - \gamma^2 \frac{\sin^2 \phi_{ij} \cos \phi_{ij}}{\rho^2} \right].
$$

We now consider the original term in question:

$$
\hat{z}_i'^\top \hat{z}_j' - \hat{z}_i^\top \hat{z}_j = \frac{1}{\rho'^2} \left[ z_i^\top z_j + 2\gamma \sin^2 \phi_{ij} - \gamma^2 \frac{\sin^2 \phi_{ij} \cos \phi_{ij}}{\rho^2} \right] - \frac{z_i^\top z_j}{\rho^2}
$$

$$
\leq \frac{1}{\rho^2} \left[ z_i^\top z_j + 2\gamma \sin^2 \phi_{ij} - \gamma^2 \frac{\sin^2 \phi_{ij} \cos \phi_{ij}}{\rho^2} \right] - \frac{z_i^\top z_j}{\rho^2}
$$

$$
= \frac{1}{\rho^2} \left[ 2\gamma \sin^2 \phi_{ij} - \gamma^2 \frac{\sin^2 \phi_{ij} \cos \phi_{ij}}{\rho^2} \right]
$$

$$
= \frac{\gamma \sin^2 \phi_{ij}}{\rho^2} \left[ 2 - \frac{\gamma \cos \phi_{ij}}{\rho^2} \right]
$$

$$
\leq \frac{2\gamma \sin^2 \phi_{ij}}{\rho^2}
$$

This concludes the proof. $\qquad\square$

## B. Simulations

### B.1. Nonparametric Simulations

For the simulations in Section 4.1, we produced two datasets, $\mathbf{X}_1$ and $\mathbf{X}_2$, independently by randomly sampling points in $\mathbb{R}^{20}$ from a standard normal distribution and normalizing them to the hypersphere. The $i$-th point in dataset $\mathbf{X}_1$ is the

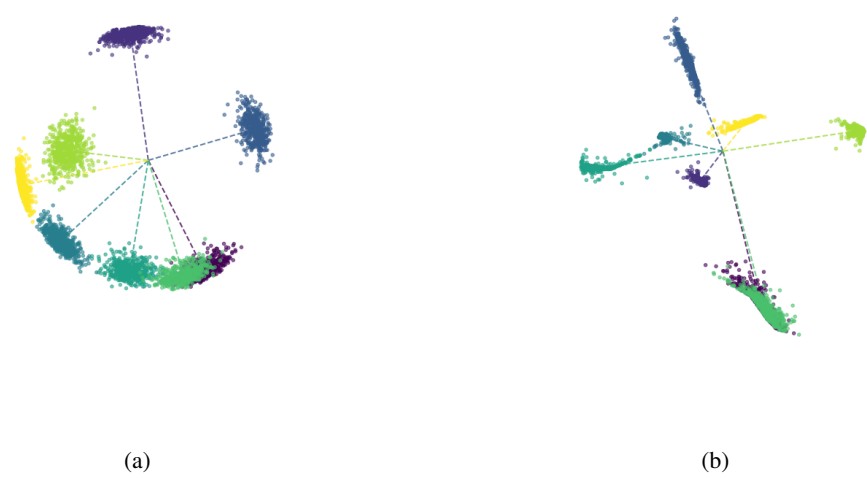

(a)                                           (b)

*Figure S2. Left*: A depiction of 8 latent classes in 3D obtained via the description in Section B.2. Dashed lines represent vectors from the origin to the mean of the distribution. *Right*: A depiction of the learned latent space (unnormalized) using the supervised InfoNCE loss after 50 epochs of training.

positive counterpart for the $i$-th point in dataset $\mathbf{X}_2$. The first dataset is then set to be static while the second is modified in order to control for the embedding norms and angles between positive pairs.

We optimize the cosine similarity by performing standard gradient descent on the embeddings themselves with learning rate 10. We consider a dataset "converged" when the average cosine similarity between positive pairs exceeds 0.999.

**Controlling for angles.** In order to control for the angle between positive pairs, we use an interpolation value $\alpha \in [-1, 1]$. Let $x_1$ be a static embedding in $\mathbf{X}_1$ and $x_2$ be the embedding in $\mathbf{X}_2$ whose angle we wish to control. In expectation, $\phi(x_1, x_2)$ will be $\pi/2$. We therefore define the embedding $x_2$ whose angle has been controlled as

$$x_2' = x_2 \cdot (1 - |\alpha|) + x_1 \cdot \alpha.$$

Specifically, the mean angle among positive pairs is controlled using the $\alpha$ parameter. In essence, when $\alpha = 0$, $x_2' = x_2$. However, when $\alpha = 1$ (resp. $\alpha = -1$), $x_2' = x_1$ (resp. $x_2' = -x_1$).

**Controlling for embedding norms.** This setting is simpler than the angles between positive pairs. We simply scale $\mathbf{X}_2$ by the desired value.

### B.2. Parametric Simulations

We restate the entire implementation for the simulations in Section 4.2 for completeness. We choose centers for 4 latent classes $\{c_1, c_2, c_3, c_4\}$ uniformly at random from $\mathbb{S}^{10}$ by randomly sampling vectors from a standard multivariate normal distribution and normalizing them to the hypersphere. We then obtain the latent samples $\tilde{z}$ around center $c_i$ via $z \sim \mathcal{N}(c_i, 0.1 \cdot \mathbf{I})$ and re-normalizing to the hypersphere. For each center, we produce 1000 latent samples; these constitute our latent classes. We depict an example of 8 such latent classes (in 3 dimensions) in Figure S2a. We finally obtain the dataset by generating a random matrix in $\mathbb{R}^{11 \times 64}$ and applying it to the latent samples.

We train the InfoNCE loss via a 2-layer feedforward neural network with the ReLU activation function in the hidden layer. The network's output dimensionality is $\mathbb{R}^{11}$ so that, after normalization, it can reconstruct the original latent classes. We train the network using the supervised InfoNCE loss with a batch size of 128. Each data point's positive pair is simply another data point from the same latent class.

We visualize the learned (unnormalized) embedding space in Figure S2b.

# C. Further Discussion and Experiments

## C.1. Experimental Setup

Unless otherwise stated, we use a ResNet-50 backbone (He et al., 2016) and the default settings outlined in the SimCLR (Chen et al., 2020a) and SimSiam (Chen & He, 2021) papers. We use 1e-6 as the default SimCLR weight decay and 5e-4 as the default SimSiam one. When running on Cifar-10 and Cifar-100, we amend the backbone network's first layer as detailed in Chen et al. (2020a). We use embedding dimensionality 256 in SimCLR and 2048 in SimSiam. When reporting embedding norms, we use the projector's output in SimCLR and the predictor's output in SimSiam: these are the spaces where the loss function is applied and therefore where our theory holds.

Due to computational constraints, we run with batch-size 256 in SimCLR. Although each batch is still 256 samples in SimSiam, we simulate larger batch sizes using gradient accumulation. Thus, our default batch-size for SimSiam is 1024. Our base learning rate is set to $0.18 \cdot \text{BatchSize}/256$ for SimCLR and $0.12 \cdot \text{BatchSize}/256$ for SimSiam. We employ a 10-epoch linear warmup followed by cosine scaling. For all cases, we use $k = 200$ for the $k$-NN classifier and apply it on the normalized embeddings.

## C.2. Opposite-Halves Effects

We devote this section of the Appendix to studying the role of the angle between positive samples on the cosine similarity's convergence under gradient descent. Referring back to Figure 2a, we see that the effect is most impactful when the angle between positive embeddings is close to $\pi$, i.e. $\phi_{ij} > \pi - \varepsilon$ for $\varepsilon \to 0$. The following result shows that this is exceedingly unlikely for a single pair of embeddings in high-dimensional space:

**Proposition C.1.** *Let* $x_i, x_j \sim \mathcal{N}(0, \mathbf{I})$ *be d-dimensional, i.i.d. random variables and let* $0 < \varepsilon < 1$. *Then*

$$\mathbb{P}\left[\hat{x}_i^\top \hat{x}_j \geq 1 - \varepsilon\right] \leq \frac{1}{2d(1-\varepsilon)^2}. \tag{7}$$

*Proof.* By Smith et al. (2023), the cosine similarity between two i.i.d. random variables drawn from $\mathcal{N}(0, \mathbf{I})$ has expected value $\mu = 0$ and variance $\sigma^2 = 1/d$, where $d$ is the dimensionality of the space. We therefore plug these into Chebyshev's inequality:

$$\Pr\left[\left|\frac{x_i^\top x_j}{\|x_i\| \cdot \|x_j\|} - \mu\right| \geq k\sigma\right] \leq \frac{1}{k^2}$$

$$\to \Pr\left[\left|\frac{x_i^\top x_j}{\|x_i\| \cdot \|x_j\|}\right| \geq \frac{k}{\sqrt{d}}\right] \leq \frac{1}{k^2}$$

We now choose $k = \sqrt{d}(1 - \varepsilon)$, giving us

$$\mathbb{P}\left[\left|\frac{x_i^\top x_j}{\|x_i\| \cdot \|x_j\|}\right| \geq 1 - \varepsilon\right] \leq \frac{1}{d(1-\varepsilon)^2}.$$

It remains to remove the absolute values around the cosine similarity. Since the cosine similarity is symmetric around 0, the likelihood that its absolute value exceeds $1 - \varepsilon$ is twice the likelihood that its value exceeds $1 - \varepsilon$, concluding the proof.

We note that this is actually an extremely optimistic bound since we have not taken into account the fact that the maximum of the cosine similarity is 1. $\qquad\square$

The above proposition represents the likelihood that *one* pair of embeddings has large angle between them. It is therefore *exponentially* unlikely for every pair of embeddings in a dataset to have angle close to $\pi$, as we would require Proposition C.1 to hold across every pair of embeddings. Thus, the opposite-halves effect is exceedingly unlikely to occur.

In accordance with this, Table S1 shows that, after one epoch of training, positive pairs of embeddings have angle greater than $\pi/2$ at a rate of around $5\%$ and $25\%$ for SimCLR and SimSiam/BYOL, respectively. So even if the 'strongest' variant of the opposite-halves effect is not occurring, a weaker one may still be. However, very early into training (epoch 16), every method has a rate of effectively 0 for the opposite-halves effect. Furthermore, the rates in Table S1 measure how

*Table S1.* The rate at which embeddings are on opposite sides of the latent space (angle between a positive pair is greater than $\pi/2$) for various datasets and SSL models.

| Model | Dataset | Effect Rate Epoch 1 | Effect Rate Epoch 16 |
|---|---|---|---|
| SimCLR | Imagenet-100 | 2% | 0% |
| | Cifar-100 | 11% | 1% |
| SimSiam | Imagenet-100 | 26% | 1% |
| | Cifar-100 | 21% | 0% |
| BYOL | Imagenet-100 | 28% | 1% |
| | Cifar-100 | 20% | 0% |

*Table S2.* $k$NN accuracies for SimSiam trained with various batch sizes. We performed training for both the default and cut-initialized variants and reported $k$NN accuracies at 100 and 500 epochs.

| Epoch | | Batch Size | | |
|---|---|---|---|---|
| | | 256 | 512 | 1024 |
| 100 | Default | 46.1 | 41.2 | 32.6 |
| | Cut ($c = 9$) | 43.1 | 46.5 | 44.3 |
| 500 | Default | 59.1 | 60.4 | 61.3 |
| | Cut ($c = 9$) | 59.4 | 58.9 | 61.5 |

*Table S3.* Linear probe accuracies at epoch 500 for default, cut-initialized and GradScale training on a subset of our image datasets.

| | | Cifar-100 | Tiny Imagenet |
|---|---|---|---|
| SimCLR | Default | 59.8 | 41.9 |
| | Cut ($c = 3$) | 63.2 | 42.8 |
| | GradScale | 62.2 | 43.2 |
| SimSiam | Default | 63.7 | – |
| | Cut ($c = 9$) | 64.2 | – |

*Table S4.* Final embedding norms at epoch 100 for SimCLR trained with Adam optimizer on Cifar-10, demonstrating the effect of different mitigation strategies on embedding magnitude.

| Method | Embedding Norm |
|---|---|
| Default | 81.0 |
| Cut | 2.1 |
| GradScale | 174.8 |

often $\phi_{ij} > \frac{\pi}{2}$. This is the absolute weakest version of the opposite-halves effect. Thus, while some weak variant of the opposite-halves effect may occur at the beginning of training, it does not have a strong impact on the convergence dynamics and, in either case, disappears quite quickly.

### C.3. Weight Decay

We evaluate the effect of weight decay in the imbalanced setting in Figure S3, which is an analog of Figure 5 for the imbalanced Cifar-10 dataset detailed in Section 6. We again see that using weight decay controls for the embedding norms and improves the convergence of both models. In correspondence with the other results on imbalanced training, we find that stronger control over the embedding norms leads to improved convergence: the high weight decay value does not perform as poorly on SimCLR as in Figure 5 and, on SimSiam, outperforms the other weight decay options.

### C.4. Cut-Initialization

We plot the effect of the cut constant on the embedding norms and accuracies over training in Figure S4. To make the effect more apparent, we use weight-decay $\lambda = 5e - 4$ in all models. We see that dividing the network's weights by $c > 1$ leads to immediate convergence improvements in all models. Furthermore, this effect degrades gracefully: as $c > 1$ becomes $c < 1$, the embeddings stay large for longer and, as a result, the convergence is slower. We also see that cut-initialization has a more pronounced effect in attraction-only models – a trend that remains consistent throughout the experiments.

We also show the relationship between cut-initialization and the network's batch size on SimSiam in Table S2. Consistent with the literature, we see that training with large batches provides improvements to training accuracy. However, we note that larger batch sizes also significantly slow down convergence. However, cut-initialization seems to counteract this and accelerate convergence accordingly. Thus, training with cut-initialization and large batches seems to be the most effective method for SSL training (at least in the non-contrastive setting).

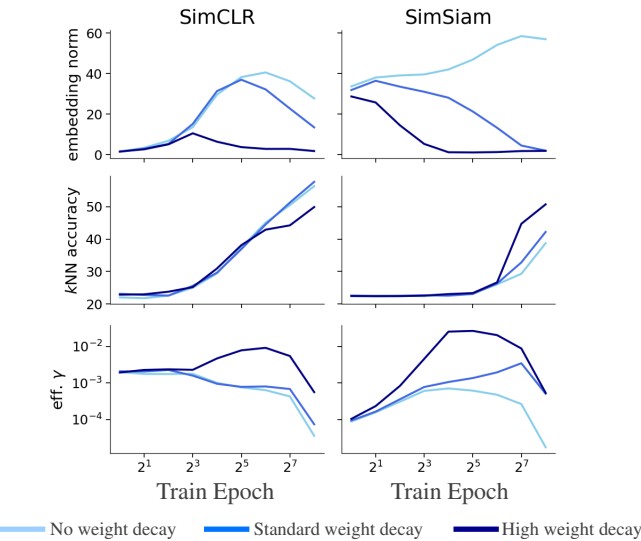

*Figure S3.* An analog to Figure 5 performed on the exponentially imbalanced Cifar-10 dataset. Weight decays are [0, 1e-5, 5e-2] for SimCLR and [0, 5e-4, 5e-2] for SimSiam. We plot the effective learning rate in the bottom row. This is calculated by scaling the learning rate by the inverse of the mean embedding norm.

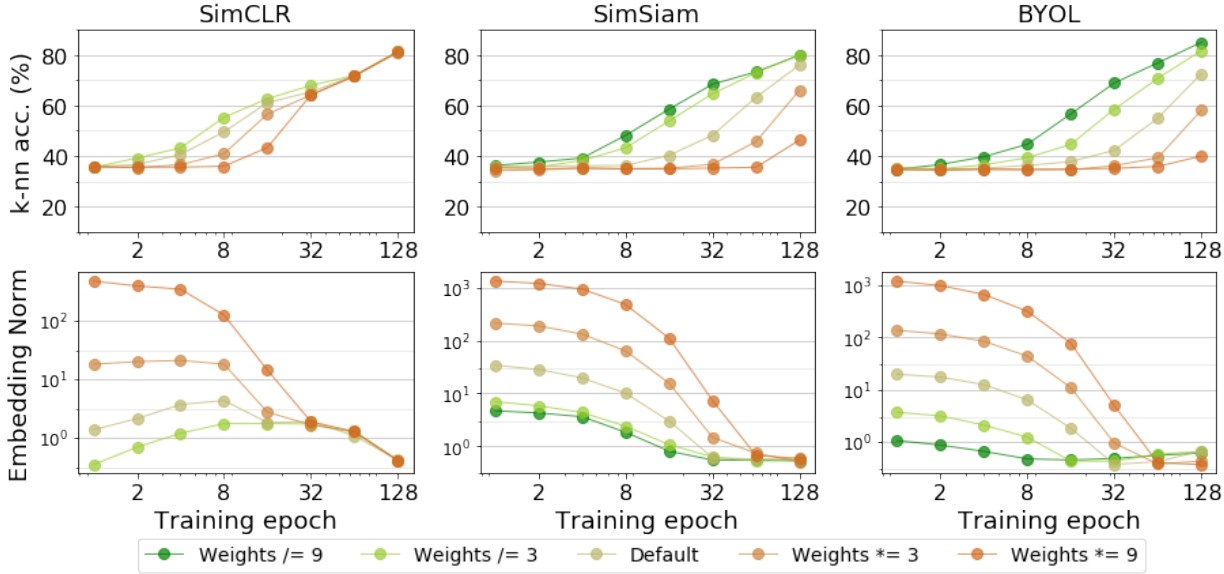

*Figure S4.* The effect of cut-initialization on Cifar-10 SSL representations. $x$-axis and embedding norm's $y$-axis are log-scale. $\lambda = 5e-4$ in all experiments.

## D. More details on gradient scaling layer

An implementation of our GradScale layer can be found in Listing 2 and pseudocode in Algorithm 1. We note that this layer is purely a PyTorch optimization trick and does not amount to implicitly choosing a different loss function:

**Proposition D.1.** *Let $t \in \mathbb{R}^n$ be a unit vector, $p : \mathbb{R}^n \backslash \{0\} \to [-1, 1], z \mapsto t^\top z/\|z\|$ the cosine similarity with respect to $t$, $\alpha \in \mathbb{R}$, and $\sigma : \mathbb{R}^n \to \mathbb{R}, z \mapsto \|z\|^\alpha$. Then the vector field $\sigma \nabla p$ has a potential $q$, i.e., $\nabla q = \sigma \nabla p$, only for $\alpha = 0$.*

*Proof.* Suppose $\sigma \nabla p$ has potential. Consider two paths with segments $s_1, s_2$ and $s_3, s_4$ going $t \to 2t \to -2t$ and $t \to -t \to -2t$, where the segments $s_1, s_4$ scaling $\pm t \to \pm 2t$ are straight lines and the other segments $s_2, s_3$ follow great

*Listing 2.* PyTorch code for gradient scaling layer

```python
class scale_grad_by_norm(torch.autograd.Function):
    @staticmethod
    def forward(ctx, z, power=0):
        ctx.save_for_backward(z)
        ctx.power = power
        return z
    @staticmethod
    def backward(ctx, grad_output):
        z = ctx.saved_tensors[0]
        power = ctx.power
        norm = torch.linalg.vector_norm(z, dim=-1, keepdim=True)
        return grad_output * norm**power, None
```

---

**Algorithm 1** Pytorch-like pseudo-code using the gradient scaling layer

---

**Input:** Encoder network $model$, gradient scaling power $p$

$z = model(batch)$

$z = grad\_scaling\_layer.apply(z, p)$

$sim = (\frac{z}{\|z\|})^T \frac{z}{\|z\|}$

$loss = InfoNCE(sim)$

$loss.backward()$

---

circles on $S^{n-1}$. By Proposition 3.1, we know that $\nabla p(z) = 0$ for $z \in \mathbb{R}_{\neq 0} \cdot t$. So $\sigma \nabla p$ is zero on $s_1$ and $s_4$. Moreover, we have

$$\int_{s_2} \sigma \nabla p \, dz = \int_{s_2} \|z\|^\alpha \nabla p \, dz = \int_{s_2} 2^\alpha \nabla p \, dz = 2^\alpha \int_{s_2} \nabla p \, dz = 2^\alpha \big( p(2t) - p(-2t) \big) = 2^{\alpha+1} \tag{8}$$

and similarly

$$\int_{s_3} \sigma \nabla p \, dz = 1^\alpha \cdot 2 = 2. \tag{9}$$

Since we assume the existence of a potential, we can use path independence to conclude

$$2^{\alpha+1} = \int_{s_2} \sigma \nabla p \, dz = \int_{s_1, s_2} \sigma \nabla p \, dz = \int_{s_3, s_4} \sigma \nabla p \, dz = \int_{s_3} \sigma \nabla p \, dz = 2. \tag{10}$$

Thus, $\alpha = 0$ and $\sigma$ does not perform any scaling. $\qquad\square$

## E. Additional figures

We provide a bar plot analogous to Figure 3 in Figure S5.

We also show each Cifar-10 class's 10 highest and 10 lowest embedding-norm samples in Figure S6. These are obtained after training default SimCLR on Cifar-10 for 512 epochs. We see that the high-norm class representatives are prototypical examples of the class while the low-norm representatives are obscure and qualitatively difficult to identify. This property was originally shown by Kirchhof et al. (2022).

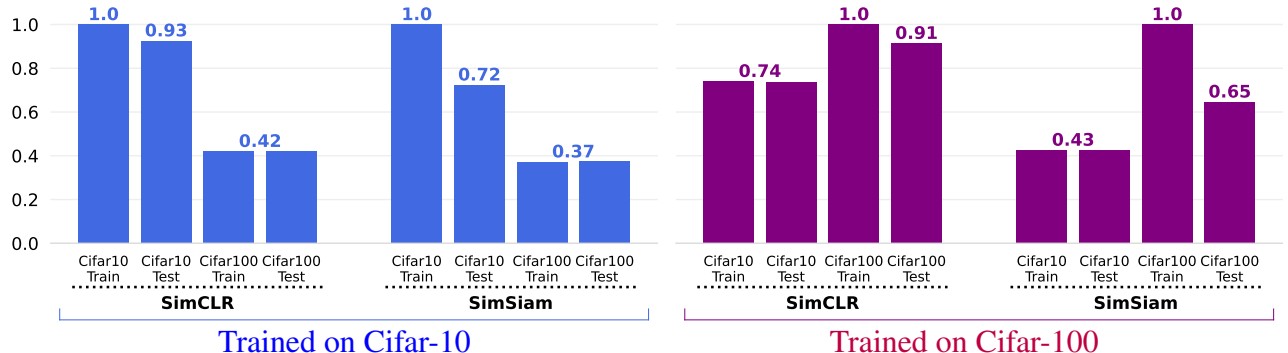

*Figure S5.* Bar plot which is analogous to Figure 3 showing embedding magnitudes on each dataset split as a function of which dataset the model was trained on. All values are normalized by training set's mean embedding magnitude. Normalized means are represented by black bars. We use the same data augmentations for the train and test sets for consistency.

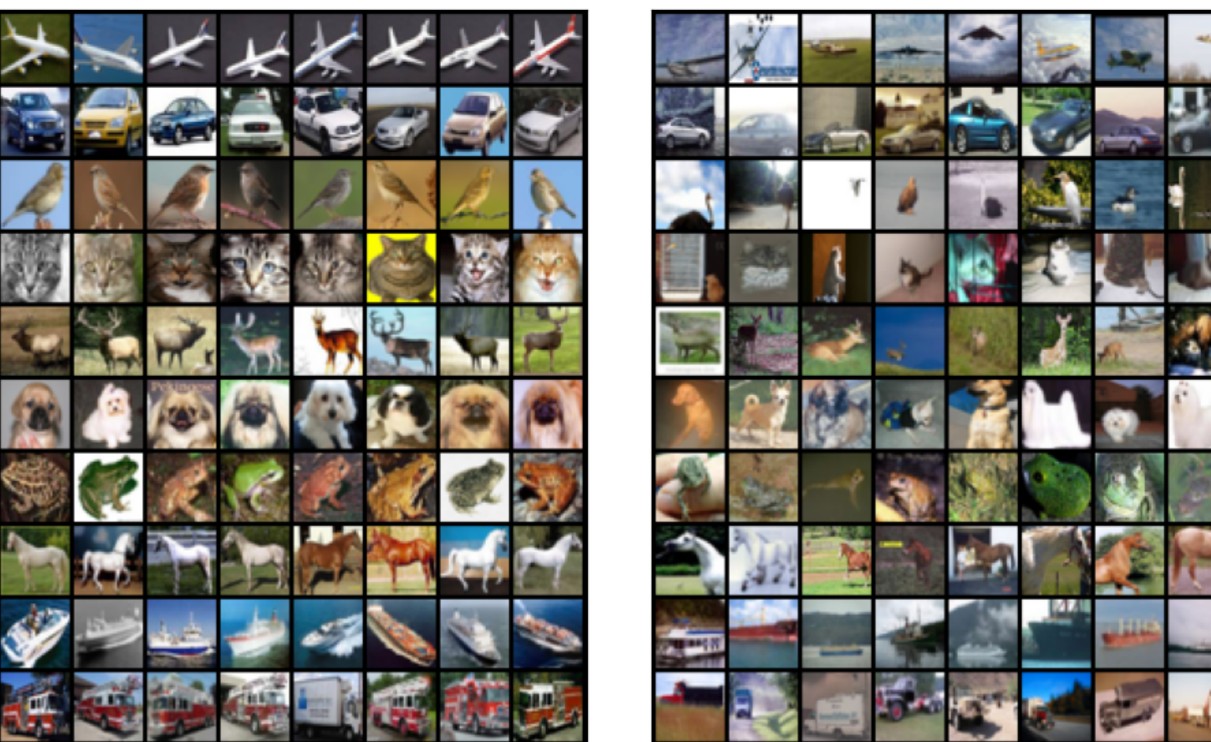

*Figure S6. Left*: Highest-norm representatives (top 10) per class. *Right*: Lowest-norm representatives (bottom 10) per class. All from default SimCLR trained on Cifar-10.

