# OpenReview forum: "On the Importance of Embedding Norms in Self-Supervised Learning"
_ICML.cc/2025/Conference — ICML 2025 poster_

### Official Review · Reviewer_8y1U · 2025-02-27

**Overall Recommendation:** 2

**Summary:**

This paper explores the role of embedding norms in self-supervised learning (SSL). It shows that embedding norms are crucial for SSL models in two key aspects: they influence convergence rates and encode model confidence. The study demonstrates that smaller embedding norms are associated with unexpected samples, and manipulating embedding norms can affect training efficiency. The paper includes theoretical analysis alongside simulations and experiments to establish how embedding norms influence the dynamics of SSL training. The results reveal that embedding norms govern how SSL models train and evolve, offering insights into the optimization process and network behavior. The study highlights the importance of controlling these norms for better performance in SSL tasks.

**Claims And Evidence:**

The claims in this paper are generally supported by evidence, but the study's limited scope—focused on just a few SSL methods (SimCLR and SimSiam)—means that these findings may not universally apply to other SSL methods. This is not explicitly stated in the paper, which could lead to questions about the generalizability of the results. Therefore, while the empirical evidence is convincing, a broader range of methods and further analysis could provide stronger and more universally applicable results.

**Essential References Not Discussed:**

The author emphasizes the importance of Embedding Norm in self-supervised learning. However, it is worth noting that methods like Barlow Twins[1] and VICReg[2] do not adopt Embedding Norm; rather, incorporating it leads to collapse in these methods. The author should delve deeper into analyzing these phenomena.

[1] Self-supervised learning via redundancy reduction. In ICML, 2021.

[2] Vicreg: Varianceinvariance-covariance regularization for self-supervised learning. In ICLR, 2022.

**Experimental Designs Or Analyses:**

1. The experiments are conducted on CIFAR-10, CIFAR-100, and ImageNet-100, which are widely accepted SSL benchmarks. The models are trained for 128–256 epochs, with SimCLR and SimSiam used as the primary SSL models. However, standard benchmarks[1] require 1000 epochs on CIFAR-10/100 and 400 epochs on ImageNet-100 for proper convergence.The limited training time raises concerns about whether the observed effects persist in fully trained models.

2. The paper only evaluates a few SSL methods, specifically SimCLR and SimSiam, while omitting other prominent frameworks like Barlow Twins, VICReg, SwAV, and DINO. Since Barlow Twins and VICReg do not use embedding norms and actually experience collapse when norms are introduced, it is crucial to analyze their behavior to confirm whether the findings hold across SSL paradigms. The lack of diversity in SSL models weakens the generalizability of the study.

[1] solo-learn: A library of self-supervised methods for visual representation learning.

**Methods And Evaluation Criteria:**

The paper uses both theoretical analysis and empirical experiments, which are appropriate for exploring the relationship between embedding norms and SSL dynamics. The theoretical component provides mathematical bounds and insights into how norms influence convergence rates and gradient magnitudes, while the empirical experiments validate these claims under practical conditions.

**Other Comments Or Suggestions:**

See Other Strengths And Weaknesses.

**Other Strengths And Weaknesses:**

Significance:

The insights into embedding norm manipulation could have important implications for optimizing SSL models, especially in terms of convergence efficiency. By providing methods like cut-initialization and weight decay to control norms, the paper offers practical techniques that could be applied to a wide range of self-supervised methods to improve training speed and stability. This is a significant contribution to the SSL field, as improving training dynamics remains a crucial challenge in modern machine learning.

Weaknesses:
1. Limited Scope of Methods: while the paper makes a significant contribution to understanding the role of embedding norms in SSL, the study is limited to just two SSL methods: SimCLR and SimSiam. These models represent only a small subset of the diverse family of SSL techniques. The paper does not discuss or evaluate methods like Barlow Twins, VICReg, or DINO, which could have provided a more comprehensive analysis. Including these methods would have made the findings more generalizable and would have strengthened the argument that the observed effects of embedding norms are consistent across different SSL paradigms.

2. Theoretical Depth: the theoretical contributions, while useful, lack novel insights and mostly build on prior work, especially in terms of embedding norm behavior. The mathematical analysis of how norms affect convergence rates and gradient scaling is solid, but the lack of original theoretical contributions (such as new theorems or proofs) limits the novelty of the paper’s theoretical aspect. The paper could have been more impactful if it had introduced new theoretical findings that directly contribute to advancing the understanding of norm manipulation in SSL.

3. Training Epochs: the training duration used in the experiments (128–256 epochs) is relatively short, especially for datasets like CIFAR-10/100 and ImageNet-100, which typically require much longer training times (e.g., 1000 epochs for CIFAR-10/100). This raises concerns about whether the effects of embedding norm manipulation hold during the full training process or if they only manifest after longer periods of training. A more extended evaluation would provide more robust evidence of the claims, particularly regarding long-term stability and model performance.

**Questions For Authors:**

I am curious whether the same phenomenon would occur with embedding norm in clustering and whitening-based methods, such as SwAV and W-MSE.

I would be happy to raise the score if authors' rebuttal can solve my concerns.

**Relation To Broader Scientific Literature:**

The contributions of this paper are rooted in and extend the broader literature on embedding norms in SSL. By mainly offering new empirical results, and practical techniques for manipulating embedding norms, the paper provides contributions to the understanding of SSL dynamics and introduces methods to improve the training and performance of SSL models. The work builds on and extends prior findings while addressing gaps in the literature related to convergence speed, confidence encoding, and embedding norm manipulation.

**Theoretical Claims:**

I have checked the correctness of proofs in this paper. However, most theoretical analyses in this paper build on prior work.

In addition, the theorem 3.4 is easy but unrealistic in my opinion, as it only involves the cosine similarity of positive pairs and neglects the interactions of negative samples which are the important part to avoid feature collapse.

As a result, the proofs lacks theoretical novelty.

---

> ### Author Rebuttal · Authors · 2025-03-31
>
> Thank you for the helpful suggestions and the in-depth review! We include experiments and discussion in response to your questions on the SSL methods used in the paper, the training epochs and the applicability/novelty of our analysis. Responses to individual questions are below:
>
> > "while the paper makes a significant contribution to understanding the role of embedding norms in SSL, the study is limited to just two SSL methods"
>
> We agree with the reviewer that adding more methods would be beneficial. To address this concern, we have run additional experiments on BYOL (resnet backbone; optimizes the cosine similarity), Mocov3 (ViT backbone; optimizes InfoNCE) and Dino (ViT backbone; not cos.sim.-based) on ImageNet-100. The results can be found in the response to Reviewer FWxF and are in accordance with the results in the paper. Please let us know if this has helped to address the concern.
>
> > "I am curious whether the same phenomenon would occur with embedding norm in clustering and whitening-based methods, such as SwAV and W-MSE."
>
> Our key theoretical contribution (norms grow, gradients decrease by norm) holds generally for loss functions based on normalized embeddings. As a result, it applies to SwAV, as well as the similar ProtoNCE, and W-MSE. The latter two directly use the cosine similarity, so Prop. 3.1 and Thm. 3.4 apply directly. Additional methods which use normalized embeddings for which our results apply include NN-CLR, ProtoNCE, BYOL, CLIP, DCL, and BEITv2, to name a few. We will make this clearer when revising the paper and will include a formalization of the general statement (that our findings hold for loss functions which depend on normalized embeddings).
>
> > "Methods like Barlow Twins[1] and VICReg[2] do not adopt Embedding Norm; incorporating it leads to collapse in these methods."
>
> We agree that this is an interesting direction for future work. As the reviewer points out, several SSL models perform worse when the embeddings are normalized. However, our analysis does not apply to these methods and we are not able to test why Barlow Twins and VICREG fail in the normalized setting in the limited rebuttal period. We will clarify this during the revision.
>
> > "theorem 3.4 is unrealistic in my opinion, as it only involves the cosine similarity of positive pairs and neglects the interactions of negative samples"; "the theoretical contributions, while useful, lack novel insights"
>
> We maintain that theorem 3.4 is a novel theoretical contribution and note that it holds immediately for the non-contrastive models SimSiam and BYOL, both of which only optimize the cosine similarity between positive samples. While it does not directly apply to the InfoNCE loss, we believe that our extensive analysis in Section 6 describes how embedding norms and convergence interact in contrastive settings. We also refer to our rebuttal to reviewer Jmax, where we extended the simulations in Section 4.1 to further analyze how weight-decay affects convergence.
>
> > "The lack of original theoretical contributions (such as new theorems or proofs) limits the novelty of the paper’s theoretical aspect."
>
> We recognize that our theoretical contributions build upon existing work in deep metric learning. However, our paper's novelty lies not in creating new mathematical frameworks, but in bringing theory and observations together and showing that they are causally related. The theory about gradient norms had not previously been experimentally verified and the experimental observations about SSL confidence did not have a theoretic underpinning. Our theory, simulations and experiments were designed to validate and explore these connections.
>
> > "the training duration used in the experiments (128–256 epochs) is relatively short"
>
> To address this concern, we have run on Cifar10/100 for 1000 epochs and report the kNN accuracies below:
>
> |SimCLR|Default|Cut|GradScale|
> |---|---|---|---|
> |Cifar10|87.7|88.2|88.2|
> |Cifar100|56.6|60.1|58.2|
>
> |SimSiam|Default|Cut|
> |---|---|---|
> |Cifar10|88.1|88.4|
> |Cifar100|61.8|62.6|
>
> The performance improvements of cut-initialization and GradScale persist after 1000 epoch training and even grow in the case of Cifar100. We thank the reviewer for the suggestion.
>
> The ImageNet experiments in Tables 2 and 3 were already run for the requested 500 epochs.
>
> ------
>
> We hope we've addressed your concerns but please let us know if there is something else we can do to convince you further.

---

### Official Review · Reviewer_wGR1 · 2025-03-02

**Overall Recommendation:** 4

**Summary:**

The paper proposes that the norm of the embeddings play an important role that may affect both optimization (convergence) and generalization properties of self-supervised learning methods. The paper makes an analytical observation about how embedding norm can slow down convergence. The paper argues about a model's confidence on seen/unseen data can be linked to embedding norm empirically. The paper uses SimCLR as an example of contrastive method and SimSiam as an example of non-constrastive method in its (empirical) analysis.

**Claims And Evidence:**

Convergence: The paper makes a claim on convergence and supports it via analysis (P 3.1, P3.2, C3.3, T3.4) and uses a toy setting for empirical support. Note that P3.1 is a result from prior work (clearly noted in the paper) while the other proprositions/theorems are new work proposed in the paper (IMO)

Seen/Unseen Data: The paper uses an empirical approach to argue about embedding norm and seen/unseen data. A problem that I see  with the analysis is that the paper uses norm values that aren't clearly interpretable. I find it hard to figure out what constitutes a small norm vs big norm value

**Essential References Not Discussed:**

One area that's a significant concern is the lack of discussion on the uses of embedding rank and relationship to downstream performance. See the following works:

- \alpha-Req: https://openreview.net/forum?id=ii9X4vtZGTZ
- RankMe: https://arxiv.org/abs/2210.02885
- LiDAR: https://arxiv.org/abs/2312.04000
- CLID: https://openreview.net/forum?id=BxdrpnRHNh

Specifically, I believe the authors should check the conclusions CLID paper makes wrt kNN classifiers. Ideally, all of the above papers should be discussed by the authors as they appear to be related work. Rank is different from embedding norm but the fact that rank has been shown to correlate well with downstream performance appears to be a related part WRT generalization.

**Experimental Designs Or Analyses:**

- I found the lack of analysis/experiments on ImageNet a big concern. This is especially concerning as the newly proposed methods/interventions need to be tested at least at ImageNet-1K scale as is standard practice in current SSL literature

- Additionally, the use of embedding norm makes it harder for this reader to interpret (why is 0.74 considered a god indicator of "OOD" in Figure 3 on the right)

- While kNN classifier is a good probe, I have seen linear probe and more recently attentive probe being favored in SSL literature. This omission is a concern for this reader/reviewer

- While SimCLR and SimSiam are good prototypes for SSL there are other recent methods like I-JEPA, DinoV2 that needs to be considered if the authors are interested in testing non-contrastive methods. While I understand that jumping off point is InfoNCE loss analysis, the authors already consider SimSiam so using popular SSL methods would make the paper interesting to readers as well as the authors.

**Methods And Evaluation Criteria:**

The paper uses a mix of analysis, toy models and small-scale experiments. Datasets include CIFAR-10, CIFAR-100 and ImageNet-100 (a subset of ImageNet). The paper uses SimSiam and SimCLR as stated earlier for as model prototypes.

While the above is good, I found the lack of analysis/experiments on ImageNet a big concern. This is especially concerning as the newly proposed methods/interventions need to be tested at least at ImageNet-1K scale as is standard practice in current SSL literature

**Other Comments Or Suggestions:**

- Figure 3 could be improved
  - (a) the axes would reader better if transposed
  - (b) the colors make it hard to distinguish classes

**Other Strengths And Weaknesses:**

- The paper is well written. I really mean this as this has been an enjoyable read
A weakness I see is that the new interventions (Cut-Initialization and GradScale) needs thorough testing at a larger scale than what the authors used to ensure the observations may be useful to a broader group of readers in SSL

**Questions For Authors:**

Please see my comments and observations made above. I look forward to authors' response as that may help clarify my questions and/or confusion about the work reported in the paper.

## Updated score to indicate my support for the paper

**Relation To Broader Scientific Literature:**

One area that's a significant concern is the lack of discussion on the uses of embedding rank and relationship to downstream performance. See the following works:

- \alpha-Req: https://openreview.net/forum?id=ii9X4vtZGTZ
- RankMe: https://arxiv.org/abs/2210.02885
- LiDAR: https://arxiv.org/abs/2312.04000
- CLID: https://openreview.net/forum?id=BxdrpnRHNh

(see section on essential references on suggested changes to manuscript)

**Theoretical Claims:**

The theoretical claims appears to be fine. I checked P3.1, P3.2, C3.3 and T3.4 and read/skimmed proofs in the appendix

---

> ### Author Rebuttal · Authors · 2025-03-31
>
> Thank you for the extensive analysis and suggestions for how to improve our work! Among other things, our rebuttal includes experiments on the Tiny-ImageNet dataset, on additional models, additional probes and on how the embedding's rank corresponds to the embedding norm. We respond to individual points below:
>
> > "I found the lack of analysis/experiments on **ImageNet** a concern."
>
> Unfortunately, due to computational constraints, this is infeasible for us. However, to address this concern, we trained SimCLR for 500 epochs on the Tiny-Imagenet dataset, which contains 200 classes, each with 500 training samples. The results are consistent with our other experiments:
>
> |Probe|Default|Cut|GradScale|
> |---|---|---|---|
> |kNN probe|36|37.7|38.1|
> |Lin. probe|41.9|42.8|43.2|
>
> We will include these when updating the paper.
>
> > "I have seen linear probe... favored in SSL literature"
>
> We focused on kNN as it is known to lower-bound the other probes [1] and is a good indicator of model performance [2]. However, to address this concern, we have also run linear probes on several datasets:
>
> **Tiny-ImageNet** linear probe results are presented in the table above.
>
> **CIFAR-100** linear probe results can be found below:
>
> |SimCLR|Default|Cut|GradScale|
> |---|---|---|---|
> |Cifar100|59.8|63.2|62.2|
>
> |SimSiam|Default|Cut|
> |---|---|---|
> |Cifar100|63.7|64.9|
>
> In both cases, the linear probe results are in line with the kNN ones. We will include linear probe evaluations in the revision.
>
> > "using popular SSL methods would make the paper interesting to readers as well as the authors"
>
> We agree that adding more methods would be beneficial. To this end, we have run additional experiments on BYOL (resnet backbone; optimizes the cosine similarity), Mocov3 (ViT backbone; optimizes InfoNCE) and Dinov2 (ViT backbone; not cos.sim.-based) on ImageNet-100. The results can be found in the response to Reviewer FWxF and are consistent with our other experiments.
>
> > "the use of embedding norm makes it harder to interpret (why is 0.74 considered a good indicator of "OOD" in Figure 3)"
>
> The absolute embedding norms will differ between models and would be difficult to use directly as a confidence metric. Instead, we propose using the *relative* embedding norms. Put simply, if an embedding norm is smaller than most norms seen on the training set, then the sample is likely OOD. Thus, in Figure 3, we have normalized the values by the training set's mean embedding norm, implying that the expected in-distribution norm is 1. The closer to 0 the value is, the more likely the sample is to be OOD. We will make this more clear in the revision.
>
> > "lack of discussion on the uses of embedding rank and relationship to downstream performance"
>
> We thank the reviewer for these references, we will include them in the related work section. Although the references address a different aspect of SSL embeddings than we do (they discuss the rank of all the embeddings whereas we discuss the norm of individual embeddings), we agree that these topics are related. Specifically, our work shows that the embedding norms grow in regions of the latent space which have high density (Section 4.2). This implies that these regions may induce large eigenvalues on the covariance matrix or be clusterable as in CLID.
>
> As a preliminary test, we include below a table which shows the rank of the Cifar10 embedding space over training epochs and the corresponding mean embedding norm. We calculate rank as the number of principal components required to capture 99 percent of the latent space's variance. Interestingly, we find that the rank starts growing at roughly the same epoch where the norms stop growing, implying a potential correlation. We will include this in the discussion section at the end of the paper.
>
> |Epoch|1|2|4|8|16|32|64|128|
> |---|---|---|---|---|---|---|---|---|
> |Rank|23|13|10|8|9|12|17|21|
> |Norm|1.4|2.5|5.5|12.9|24.1|22.7|23.5|22.1|
>
> We also note that this is likely related to the study on dimensional collapse in SSL methods, such as [3].
>
> > "the new interventions need testing at a larger scale"
>
> We emphasize that these interventions were included to study the behavior of the embedding-norm effect in practice and were not intended to obtain SOTA results. However, we agree that further testing would be helpful. To this end, we hope the analysis on additional models (BYOL, MoCov3 and Dino on ImageNet-100) and the additional experiments on Tiny-Imagenet have addressed this concern.
>
> > Figure 3
>
> We will amend the figure.
>
> ### References
>
> [1]: Oquab, Maxime, et al. "Dinov2: Learning robust visual features without supervision." TMLR 2023.
>
> [2]: Marks, Markus, et al. "A closer look at benchmarking self-supervised pre-training with image classification." arXiv preprint 2024.
>
> [3]: Tian, Yuandong, et al. "Understanding self-supervised learning dynamics without contrastive pairs." ICML 2021.
>
> ------
>
> Please let us know if these have addressed your concerns.

---

> > ### Comment · Reviewer_wGR1 · 2025-04-01
> >
> > I thank the authors for their rebuttal. The rebuttal adds good support to existing content in the paper but concerns with empirical setup/analysis remain. I will consider the authors rebuttal in my discussion with the other reviewers and AC in the next phase. I appreciate your hard work.

---

### Official Review · Reviewer_Jmax · 2025-03-08

**Overall Recommendation:** 3

**Summary:**

The article examines the structure of the gradient expression for the InfoNCE SSL objective. Building on a previous result, this gradient expression is reformulated to emphasize that:

- The gradient involves a projection onto a subspace orthogonal to the embedding vector with respect to which the gradient is computed.
- The gradient is inversely proportional to the norm of this embedding vector.

These properties are utilized to derive learning characterizations, such as the continuous growth of embedding vector norms and an upper bound on the improvement of the cosine similarity between positive pairs. These characterizations are  then applied to analyses, including the use of embedding norms for addressing class imbalance and assessing network confidence, as demonstrated through numerical examples.

**Claims And Evidence:**

- The claims regarding the special structure of the cosine distance gradient—specifically, its projection onto the tangent space and its inverse dependence on the embedding norm—are extensions of previous work and are well-supported by straightforward derivations.

- The claim about the increase in embedding norm and the dependence of cosine angle improvement is analytically grounded in the gradient structure.

- The claims concerning class imbalance and confidence are motivated by the earlier theoretical findings but are primarily supported by numerical experiments.

These claims are specific to SSL loss functions that involve cosine distance and optimization schemes that use a constant or non-adaptive learning rate.

**Essential References Not Discussed:**

I believe the article offers essential references including  Zhang et al. (2020) which provides the structural form of the  cosine distance gradient, which is extended and further elaborated in this article.

**Experimental Designs Or Analyses:**

The experimental designs seem appropriate, although the presentation clarity for Figure 2b (Section 4.2 ) and Figure 3 (Section 5) needs improvement.

**Methods And Evaluation Criteria:**

The setups for numerical experiments based on synthetic data and classical ML datasets mainly make sense as well as the conclusions drawn from these examples.

**Other Comments Or Suggestions:**

Presentation improvement suggestions: There appears to be various typos:

- Line 802: R^20->R^{20}?
- Line 895: variables->vectors
- Line 926: Table ??

**Other Strengths And Weaknesses:**

Strengths:
The article provides useful insights into the learning process of cosine-distance-based self-supervised learning (SSL), particularly focusing on embedding-norm normalization and projection onto the tangent space of the embedding vector.

Weaknesses:
The main results build upon prior work on the structure of the cosine-distance gradient. Extending this analysis to the entire InfoNCE objective is relatively trivial, as it naturally follows from the use of cosine distance for negative samples.

The analysis is specific to the use of SGD-based updates for cosine-distance-based SSL with a constant or non-adaptive learning rate. However, in practice, learning rates can be chosen adaptively—for instance, using learning rate rules that scale with the norm of the gradient—thereby eliminating the quadratic slowdown claimed in Theorem 3.4. Moreover, SSL algorithms often incorporate $\ell_2$ -normalization or regularization on embeddings or projector outputs during training, which fundamentally alters the dynamics of embedding norms.

Despite the fact that  the article provides interesting insights about the impact of embedding-norm and tangent-space projection components of the  cosine-distance based SSL gradient, the overall contribution is weakened by multiple factors. Most notably, the core analysis essentially reaffirms known results about the structure of the cosine-distance gradient, making its extension to the broader InfoNCE objective feel rather trivial. Furthermore, the reliance on a strictly SGD-based framework with a fixed learning rate neglects the practical reality of common SSL training, where adaptive learning rate schedules—and often $\ell_2$-normalization or related regularization strategies—impact the claimed quadratic slowdown feature. Therefore, while there is value in the article’s focus on embedding-norm dynamics, the limited novelty in analytical extension and practical applicability diminish the strength of the overall contribution.

**Questions For Authors:**

- Based on Equation (3), the authors claim that embedding norms cause a slowdown proportional to their magnitude. In this expression, the learning rate  $\gamma$ is a tunable algorithm parameterthat can be adjusted to compensate for this effect. Could we not simply choose $\gamma \propto \rho^2$, or use adaptive learning rate rules to prevent the  slow down? Alternatively, would it not be possible to constrain embeddings, to $S^{d-1}$ for example?

- How do regularization terms in the overall loss (such as weight decay) impact the conclusions drawn about the embedding norms based on the special structure of the cosine-distance metric?

**Relation To Broader Scientific Literature:**

The article analytically extends the structural interpretation of the cosine distance gradient and provides simulation-supported insights into SSL training.

**Theoretical Claims:**

The analytical results are mainly in Section 3 (and their proofs are in Appendix A). These are extensions of the cosine angle gradient result by Zhang et al (2020) and appear to be correct. The probability bound (Proposition C.1) in Appendix C appears to be correct.

---

> ### Author Rebuttal · Authors · 2025-03-31
>
> Thank you for your thorough review. Regarding Theorem 3.4's applicability, we appreciate your insights about potential mitigations - weight decay, scaling gradients by embedding norms, and adaptive optimizers. We note that our paper systematically analyzes how the first two affect SSL training in practice. For completeness, we include adaptive optimizer experiments below.
>
> We now respond to individual comments:
>
> > "The extension to the broader InfoNCE objective is trivial"
>
> We agree that the extension from cosine similarity to the InfoNCE objective is straightforward. However, we believe that this is a strength of our paper.
>
> The InfoNCE objective function is used in countless settings (see reply for reviewer FWxF) and our work shows that, in each of these, it has a dependence on the embedding norms. Although a few of these ideas were known in the deep metric learning literature, they had not been experimentally validated or extended to standard contrastive losses. In short, it is relevant for the SSL community to be aware of how the embedding norms interact with the InfoNCE loss function.
>
> > "the core analysis reaffirms known results about the cosine-distance gradient"
>
> Quick clarification: we believe that our core analysis lies in studying the relationship between SSL training and the embedding norms with (i) new theory (Thm 3.2 and 3.4), (ii) new experiments and (iii) new mitigation strategies. Such a comprehensive analysis was previously absent from the literature.
>
> > "Could we not... use adaptive learning rate rules to prevent the slow down"
>
> We believe the reviewer is implying that the results in Theorem 3.4 do not apply under, for example, the Adam optimizer. If so, we disagree. The gradient under Adam optimization still inversely depends on the embedding norm and, consequently, the embedding norms slow down the training process. To test this, we have trained SimCLR and SimSiam on the Cifar datasets using the Adam optimizer for 100 epochs and find consistent results:
>
> |SimCLR w/ Adam|Default|Cut|GradScale|
> |---|---|---|---|
> |Cifar10|79.6|79.9|80.5|
> |Cifar100|44.3|45.3|46.6|
>
> |SimSiam w/ Adam|Default|Cut|
> |---|---|---|
> |Cifar10|73.7|79.8|
> |Cifar100|36.4|44.9|
>
> We also note that the Adam optimizer is not common in cos.sim.-based SSL models.
>
> > "Could we not choose the learning rate proportional to the embedding norm?"
>
> We agree. In fact, this is *precisely* what our GradScale does: it multiplies the gradient on each sample by its embedding's norm. We find this improves training but can lead to instability due to a positive feedback loop which is introduced: the embedding norm grows with the magnitude of the gradient and the gradient is now multiplied by the embedding norm. We include the embedding norms from the SimCLR w/ Adam Cifar10 training runs as an example of how the norms are affected by our mitigation strategies:
>
> |Default|Cut|GradScale|
> |---|---|---|
> |81.0|2.1|174.8|
>
> > "How do regularization terms in the overall loss (such as weight decay) impact the conclusions drawn?"
>
> We politely note that we extensively analyzed how weight-decay improves SSL convergence in Section 6 and the appendix, particularly Table 1 and Figures 5, S3.
>
> Additionally, weight decay would not change the correctness of Theorem 3.4: each individual step still depends on the square of the embedding norm. Nonetheless, we see the reviewer's point that weight decay will interact with the bound in Theorem 3.4, possibly leading to faster convergence. Practically, however, it is difficult to find a regularization strength large enough to mitigate the slow-down in Theorem 3.4 without training diverging.
>
> To evaluate this, we have extended the simulation from Section 4.1 by augmenting the objective with a regularization term on the embedding norm to simulate how weight-decay affects convergence. Thus, we now optimize $\mathcal{L} = -\hat{z}_i^\top \hat{z}_j + \gamma \cdot ||z_i||_2$. The results are listed in the tables below. We find that while weight-decay can speed up convergence, the number of required steps still depends quadratically on the initial embedding norm. However, if the weight decay is made too large, convergence never occurs. This is in line with our experiments in Section 6.
>
> For $\gamma = 0.5$:
> |Initial Norm|Steps to Converge|
> |---|---|
> |1|64|
> |4|526|
> |7|1080|
>
> For $\gamma = 1$:
> |Initial Norm|Steps to Converge|
> |---|---|
> |1|49|
> |4|318|
> |7|610|
>
> For $\gamma = 10$, the simulation diverges.
>
> > "Would it be possible to constrain embeddings to the sphere?"
>
> This is something we considered: it would mean forcing the outputs onto $\mathcal{S}^{d-1}$ and may work well with the loss from Koishekenov et al. [1]. We chose to implement the GradScale layer as an alternative as it accomplishes a similar effect in principle.
>
> ### References
>
> [1]: Koishekenov, et al. "Geometric contrastive learning." VIPriors Workshop, CVPR 2023.
>
> ------
>
> Please let us know if we have addressed your concerns.

---

### Official Review · Reviewer_5PH5 · 2025-03-13

**Overall Recommendation:** 4

**Summary:**

This paper investigated the relatively overlooked area of embedding normalization in self-supervised learning (SSL), as most prior works default to using cosine similarity between embedding vectors -- which normalizes by the product of magnitude of both vectors -- and effectively projects data onto a unit hypersphere. Inspired by some empirical evidence that the pre-normalization embedding norms contain meaningful information, the authors aimed to systematically establish the embedding norm’s rule in SSL. The authors claimed that embedding norms (1) govern SSL convergence rates, and (2) encode network confidence, and validated the claim with theoretical analyses, simulations and empirical results.

The first main result is on how embedding norms affect convergence. The authors first proved theoretical bounds showing that embedding norms impose a quadratic slowdown on SSL convergence, and further validated in simulation and real experiments, demonstrating the benefit of small embedding norms. Then they showed embedding norms grow during training when cosine similarity is optimized. These two results indicated that effective and efficient training of SSL requires managing the embedding norms, and they provided methods for doing this.

The second main result is on how embedding norms encode confidence. The authors argue that since the embeddings grow with each gradient update, their norms naturally correspond to the frequency of observed latent features, and this naturally correspond to model confidence. They also provided methods for studying this and validated this claim.

**Claims And Evidence:**

Yes. The claims are clearly organized and supported by evidence.

**Essential References Not Discussed:**

The paper appears comprehensive in its referencing of relevant literature. All critical prior works directly related to embedding norms in SSL seem adequately discussed, and I did not identify any essential missing references.

**Experimental Designs Or Analyses:**

The experimental designs and analyses conducted are sound and valid. I reviewed the experimental setup for results shown in Figures 2-5 and Tables 1-3. The approach is thorough and methodologically sound.

**Methods And Evaluation Criteria:**

The proposed methods (weight decay, cut-initialization, and GradScale) are reasonable and directly address the identified embedding norm effect. Even though the cut-initialization is a little too simple and brutal, in my opinion the main contribution is the insight rather than the specific novelty of the method itself.

Evaluation criteria, such as k-NN classification accuracy on standard datasets (Cifar-10, Cifar-100, ImageNet-100, Flowers102), appropriately measure SSL representation quality. The methods and evaluation criteria used in the experiments are sensible for studying the stated problems and phenomena.

**Other Comments Or Suggestions:**

Nothing at the moment.

**Other Strengths And Weaknesses:**

Overall, I find this paper to be insightful. In my opinion, the most interesting contribution is that it highlighted how embedding norms affect convergence. Specifically, the authors demonstrated both theoretically and empirically that smaller norms are preferred, and meanwhile related that to the observation that embedding norms generally increase while optimizing for cosine similarity. The conclusion that effective and efficient SSL training relies on managing the embedding norms is very insightful.

The weakness would be the proposed methods, especially the cut-initialization, are relatively brutal and less innovative. With that said, I find the knowledge and insight brought to the community by this paper to outweigh this weakness.

**Questions For Authors:**

Great work. One minor question is, besides k-NN accuracy, have you considered other measures of latent space quality (linear probing and/or finetuning performance for classification)?

**Relation To Broader Scientific Literature:**

The paper builds effectively upon existing literature regarding embedding norms and SSL. It clearly positions itself relative to foundational works such as SimCLR, SimSiam, and prior studies on embedding norms (Wang et al., 2017; Zhang et al., 2020). It provides deeper empirical insights compared to earlier works. The paper is well situated within the broader SSL literature.

**Theoretical Claims:**

I reviewed the correctness of the main theoretical claims (Proposition 3.1, Proposition 3.2 and Theorem 3.4). The proofs seem correct to me, or at least I did not identify issues. But I am not particularly good at proofs so I wouldn't rely on this analysis.

---

> ### Author Rebuttal · Authors · 2025-03-31
>
> Thank you for the kind words regarding our paper. We respond to your questions below.
>
> > "The weakness would be the proposed methods, especially the cut-initialization, are relatively brutal and less innovative"
>
> Regarding the cut-initialization, we agree with this point and it is something we deliberated on for quite some time. However, we could not come up with an alternative method for ensuring that the embedding norms are small at initialization. An alternative option is to apply cut-initialization at only the final layer, but this performs roughly equivalently. We also note that cut-initialization seems to accomplish the desired task very well, as evidenced by Tables 1-3 in the paper.
>
> If the reviewer has suggestions as to what we may do as an alternative to cut-initialization, we would be happy to try them out.
>
> > "Besides k-NN accuracy, have you considered other measures of latent space quality?"
>
> We recognize that there are several probes that can be used. We focused on kNN as it is known to lower-bound the other probes [1] and is a good indicator of model performance [2]. To address the reviewer's question, we have run the linear probe on various models at the 500 epochs mark and see the same performance improvements:
>
> |SimCLR|Default|Cut|GradScale|
> |---|---|---|---|
> |Cifar100|59.8|63.2|62.2|
> |Tiny Imagenet|41.9|42.8|43.2|
>
> |SimSiam|Default|Cut|
> |---|---|---|
> |Cifar100|63.7|64.9|
>
> We will include linear probe evaluations in the revision of the paper.
>
> [1]: Oquab, Maxime, et al. "Dinov2: Learning robust visual features without supervision." TMLR 2023.
>
> [2]: Marks, Markus, et al. "A closer look at benchmarking self-supervised pre-training with image classification." arXiv preprint 2024.
>
> ------
>
> Please let us know if there is anything else which we can address.

---

> > ### Comment · Reviewer_5PH5 · 2025-04-02
> >
> > Many thanks to the authors for providing the additional evidence. I do not have other questions at the moment.

---

### Official Review · Reviewer_FWxF · 2025-03-14

**Overall Recommendation:** 3

**Summary:**

This submission explores the embedding norms interaction with SSL training dynamics, where cosine similarity is commonly used to map the data to a hypersphere. This paper studies the gradients of the cosine similarity loss (and the InfoNCE loss) revealing that while gradients are inversely scaled by the norm of the embedding (large norms cause vanishing gradients), norms after gradient step get larger (causing a vicious cycle). This also results in an the expression for cosine similarity change after one gradient step, showing quadratic slow down in convergence wrt the norm and angle between the pair of embeddings.

These findings are then validated via a set of simulation experiments by controlling the angle and the norm. Alternatively, the final embedding norms are themselves influenced by optimization of cosine similarity - dense regions get higher norms. This leads to authors making a conclusion (though not formally treated, but empirically demonstrated) that embedding norm is descriptive of the network confidence in the data point, potentially encoding novelty (OOD samples), downstream performance (classification accuracy), human labelers' confidence and agreement.
Manipulating the embedding norm is later studied in the form of three methods (weight decay, cut-initialization and gradscale layer), showing gains over default settings and in imbalanced datasets.

**Claims And Evidence:**

For the most part evidence is convincing. However, the 'Embedding Norm as Network Confidence' section is mainly an empirical study (although theoretically grounded via norm size dependency on the sample frequency during training for cosine similarity loss), it could benefit a lot from experiments on larger dataset like Imagenet, since CIFAR datasets are relatively simple and small. While Imagenet has 1K classes, it is possible to take subsets of classes for the purposes of the study (e.g. Figure 3). The paper uses one such subset (Imagenet-100) in Figure 4, so this should not be a problem. Adapting different subsets of Imagenet for experiments on network confidence would help make the results more convincing.

For the InfoNCE loss proof, the authors argue that negative contribution is averaged across many samples and should be smaller than the positive contribution (cosine similarity loss). I'm not sure this is always true, especially at the start of training and for complex datasets and network architectures and without assuming the network is smooth enough (similar augmented images land closer in latent than negatives).

The authors also discuss 3 ways in which this submission differs from the related work on embedding normalization for deep metric learning literature, which is very commendable. I agree with the first, however, the rest do not seem to be faithfully validated. The second - evaluating how large embedding norms affect real world training - brings us back to the choice of datasets raised earlier. The third - while weight decay and cut-init are already known methods, it is not entirely clear whether GradScaler is a novel contribution of this paper or not. If yes, there should have been better empirical confirmation of it effectiveness (now it states that the models are sensitive and GradScaler fails to demonstrate gains on larger dataset). In addition, it is not verified/discussed whether addressing embedding norm affect is more beneficial than the other techniques from DML (i.e. regularization term for variance from [1]) or if they are complimentary.

**Essential References Not Discussed:**

-

**Experimental Designs Or Analyses:**

Yes, the synthetic experiments and confidence experiments description seem valid.

**Methods And Evaluation Criteria:**

The formulated experimental setups do make sense, but it would be more convincing if Imagenet-like datasets were used (see Claims and Evidence).

**Other Comments Or Suggestions:**

When describing GradScaler, it seems p=0 and p=1 are switched. Having scalar to power 0 will eliminate the scaling effect, while p=1 leaves the norm scaling intact, for now it is written the other way around if I'm not confused.

There is also broken reference to a Table in the supplementary.

**Other Strengths And Weaknesses:**

Theoretical findings largely depend on (and restate) the results from previous work (especially [1] which they acknowledge). This is not per se a serious weakness, since the authors formalized the quadratic inverse dependence on the norm in Theorem 3.4. However, this diminishes the conceptual contribution of the work.

[1] Zhang, Dingyi, Yingming Li, and Zhongfei Zhang. "Deep metric learning with spherical embedding." Advances in Neural Information Processing Systems 33 (2020): 18772-18783.

See also earlier sections (Claims and Evidence, Relation to Broader Scientific Literature)

**Questions For Authors:**

While I understand that SimCLR, SimSiam and BYOL are picked based on their use of cosine similarity, I wonder if other MSE-based methods also exhibit similar behaviour. Have you tried similar experiments of VICReg/W-MSE?

**Relation To Broader Scientific Literature:**

The submission addresses current gaps in understanding how self-supervised representation learning (SSRL) methods work, specifically those based on contrastive learning. While there has been a track of works connecting modern SSRL methods, especially SimCLR/Contrastive Predictive Coding to deep metric learning (DML), this work seem to use results obtained in earlier DML literature to describe the training dynamics of common contrastive learning-based methods based on data augmentation and provide ground for connecting embedding norms and network confidence in cosine-similarity-based methods.

While earlier discussion on network confidence via norm in the previous literature is discussed in the related work section, it seems unfair to say there has been no explanation for this phenomenon, especially when referencing work that treats norms as concentration parameter in von Mises-Fischer-based models.

**Theoretical Claims:**

I looked through the proofs in Appendix they seem alright, but I haven't checked everything very carefully (especially since the results are obtained in previous work [1]).

---

> ### Author Rebuttal · Authors · 2025-03-31
>
> Thank you for your thoughtful and in-depth review! Below, we address your concerns by adding experiments on Tiny-ImageNet and testing additional models (BYOL, MoCo v3, Dino), which confirm the generality of our theoretical and experimental results. Furthermore, we clarify the novelty of our analysis: while some of our methods and ideas have precedent, we are the first to consolidate them and show unambiguously how the embedding norms interact with self-supervised learning. We believe that knowledge of these interactions is valuable to the ICML community.
>
> Detailed responses are below:
>
> > "[The paper] could benefit from experiments on larger dataset like Imagenet"
>
> We agree that it would be nice to run on ImageNet. However, due to computational constraints, this is infeasible for us. We therefore refer to our response to Reviewer wGR1, where we include experiments on Tiny-Imagenet. The results are consistent with what we see on other datasets.
>
> > "I wonder if **other methods** also exhibit similar behaviour."
>
> To address both this and the broader question of our results' generalizability, we have trained BYOL (which optimizes the cos. sim.), MoCo v3 (which optimizes InfoNCE) and Dino (which is not cos.sim.-based) on ImageNet-100 for 100 epochs with and without cut-init. We find that cut-init improves BYOL's accuracy at 100 epochs from 26\% to 54\% and MoCo v3's accuracy from 54\% to 58\%. These are in accordance with the results in our paper (in particular, Figure S4 from the paper). Our theory only applies to losses with normalized embeddings, thus not to Dino, VicReg or Barlow Twins. Indeed, we find that cut-init drops Dino's accuracy from 44\% to 29\%. This implies that trying to resolve the embedding norm effect in settings where it does not occur hurts performance.
> Nevertheless, many other prominent SSL methods do rely on normalized embeddings (e.g., W-MSE, NNCLR, ProtoNCE, SWAV, CLIP, DCL, and BEiTv2), highlighting how widely applicable our analysis is. We will discuss this in the revision.
>
> > "it is unfair to say there has been no explanation for network confidence via norm"
>
> While we agree Kirchhof et al. (2022) have discussed the embedding norm as a concentration parameter for the vMF distribution, they only *stated* that it behaves this way and gave brief empirical evidence of this. In fact, Kirchhof et al. reference Scott et al. (2021) who explicitly raise why this holds as an open question: ``Why does the COSINE embedding convey a confidence signal in the norm?'' Our work therefore gives a clear *explanation* for this phenomenon for the first time. Furthermore, Scott et al. was in the supervised setting.
>
> > "...weight decay and cut-init are already known methods..."
>
> While weight-decay is a known method, its standard discussion in the literature is entirely different from ours and focuses on overfitting. We only know of one reference, Wang et al. (2017), which suggests that weight decay may help with the embedding norm effect but this was not tested.
>
> Similarly, cut-init only exists in transformer architectures and was proposed to manage the attention. We are not aware of it in any convolutional architectures or for managing embedding norms. Thus, the idea of dividing the weights at initialization has not been discussed with regards to the convergence rate and in this sense is a novel contribution of our paper.
>
> Both of these are complementary to the regularization term for the variance from Zhang et al. (2020).
>
> > "It is not clear whether GradScale is a novel contribution of this paper or not."
>
> GradScale is a novel contribution, but our goal with it was not to propose a new SOTA method. Instead, we use it as a way to analyze how embedding norms affect SSL training. That is, by using GradScale we can show that removing the gradient's relationship to the embedding norm can improve generalization, especially on imbalanced datasets.
>
> > "When describing GradScale, p=0 and p=1 are switched."
>
> The $p$ parameter in GradScale is the power of the embedding norm that gets additionally multiplied to the original gradient. Thus, for $p=0$, the gradient is multiplied by $\|z\|^0 = 1$, leaving the gradient untouched.
>
> > "the authors argue that negative contribution is averaged across many samples and should be smaller than the positive contribution. I'm not sure this is true"
>
> We agree with this point and will remove this from the paper. However, we note that this has no effect on the correctness of Prop 3.2. The main point is that *all* gradients of the InfoNCE loss are orthogonal to the points they act on, thus increasing the embedding points' norm, independent of the sizes of the attractive and repulsive contributions.
>
> -------
>
> We hope we have addressed your concerns. Please let us know if there is anything we can do to convince you further.

---

### Decision · Program_Chairs · 2025-05-01

**Decision:**

Accept (poster)

**Comment:**

The paper investigates the role of network’s feature norms in some aspects of self-supervised learning specifically for the case of a loss with the cosine similarity metric. It first formally studies the feature norm’s role in learning dynamics and convergence through the lens of its gradient (update steps) and empirically verifies the results for a simulated and controlled setup, arriving at the conclusion that feature norms need be directly treated for improved training dynamics.  Then, it argues that high-norm features could be due to the occurrence frequency and therefore the norm can be a proxy measure for out-of-distribution detection or “confidence” of the model. They thoroughly examine this hypothesis. Finally, they empirically investigate some implications of these two aspects interacting during training and propose and evaluate some mitigations for the adverse effects.

Five expert reviewers evaluated the paper which initially had divided scores; with two on the reject side (ratings 2, 2) and three on the positive side (ratings 3, 3, 4).
The reviewers appreciated the relevance and importance of the study and (some of) the insights it offers regarding convergence and the connection of SSL feature norm and OoD.
The main raised issues consist of (1) the reliance on prior theoretical framework and results, (2) the purely hypothetical and empirical nature of some results which are also not examined within a large-scale benchmark (relatively small datasets, short training time, low number of SSL methods), and (3) limited novelty of the proposed mitigation techniques.

The authors provided a thorough rebuttal in which (1) they present additional results covering larger datasets, more SSL methods which are trained with more epochs, (2) they agree with the simplicity of the formal results considering the prior work but argue that the formal connection is new (and important) and that the corresponding empirical results are extensive, consistent and convincing, and (3) proposed mitigations are either entirely novel or used for a novel purpose.

All reviewers attended the rebuttal which led to an increase in rating by two reviewers with the final ratings arriving at 2,3,3,4,4.

The AC agrees with the reviewers that the study is relevant, of wide interest, and brings some interesting insights and mitigation practices. The AC also agrees that the novelty is limited in a few aspects but despite the limited novelty, the AC finds the message of the paper, on the role of feature norm in training dynamics and OoD of the main family of SSL methods, considerably more solidified than any prior work and worth dissemination. Therefore, the AC agrees with the absolute majority of the reviewers and recommends acceptance.